# From Narrow to Panoramic Vision: Attention-Guided Cold-Start Reshapes Multimodal Reasoning

**Ruilin Luo**[13*]  **Chufan Shi**[2*]  **Yizhen Zhang**[1*]  **Cheng Yang**[4]  **Songtao Jiang**[5]
**Tongkun Guan**[6]  **Ruizhe Chen**[5]  **Ruihang Chu**[1]  **Peng Wang**[3]  **Mingkun Yang**[3]
**Yujiu Yang**[1†]  **Junyang Lin**[3]  **Zhibo Yang**[3]
[1]Tsinghua University  [2]University of South California  [3]Qwen Team, Alibaba Group
[4] University of California San Diego  [5] Zhejiang University  [6] Shanghai Jiao Tong University

## Abstract

The cold-start initialization stage plays a pivotal role in training Multimodal Large Reasoning Models (MLRMs), yet its mechanisms remain insufficiently understood. To analyze this stage, we introduce the *Visual Attention Score* (VAS), an attention-based metric that quantifies how much a model attends to visual tokens. We find that reasoning performance is strongly correlated with VAS ($r = 0.9616$): models with higher VAS achieve substantially stronger multimodal reasoning. Surprisingly, multimodal cold-start fails to elevate VAS, resulting in attention distributions close to the base model, whereas text-only cold-start leads to a clear increase. We term this counter-intuitive phenomenon *Lazy Attention Localization*. To validate its causal role, we design training-free interventions that directly modulate attention allocation during inference, performance gains of 1–2% without any retraining. Building on these insights, we further propose **A**ttention-Guided **V**isual **A**nchoring and **R**eflection (AVAR), a comprehensive cold-start framework that integrates visual-anchored data synthesis, attention-guided objectives, and visual-anchored reward shaping. Applied to Qwen2.5-VL-7B, AVAR achieves an average gain of 7.0% across 7 multimodal reasoning benchmarks. Ablation studies further confirm that each component of AVAR contributes step-wise to the overall gains. The code, data, and models are available at https://github.com/lrlbbzl/Qwen-AVAR.

## 1 Introduction

Recent advances in Reinforcement Learning (RL) have significantly enhanced the reasoning capabilities of Large Language Models (LLMs) such as OpenAI o1 (Jaech et al., 2024), Qwen-Max (Team, 2025) and DeepSeek-R1 (Shao et al., 2024; Guo et al., 2025). Building upon this success, recent research has leveraged RL to construct Multimodal Large Reasoning Models (MLRMs), aiming to equip them with stronger cross-modal reasoning capabilities (Zhou et al., 2025; Yang et al., 2025d; Wei et al., 2025b; Yue et al., 2025b; Li et al., 2025; Ma et al., 2026). However, applying these techniques directly exposes a critical but underexplored stage of the training pipeline: the cold-start initialization stage that precedes the RL stage. Understanding and optimizing this stage remains a core limitation of current MLRMs.

A surprising and counter-intuitive phenomenon illustrates this limitation: A text-only cold-start yields substantial improvements for MLRMs in subsequent RL tuning, whereas multimodal cold-start provides only marginal gains (Wei et al., 2025a;b; Yue et al., 2025b). This phenomenon reveals a bottleneck in current training paradigms: MLRMs fail to leverage multimodal signals during cold-start, leading to inefficient resource use and limiting the potential of RL for multimodal reasoning. Despite its importance, this paradoxical outcome still lacks a clear quantitative explanation.

---

*Equal Contribution.

†Corresponding author. yang.yujiu@sz.tsinghua.edu.cn

To shed light on this paradox, we re-examine multimodal reasoning through the lens of attention allocation (Sec. 3). We introduce *Visual Attention Score* (VAS) to quantify how much a model attends to visual tokens. Correlating VAS with reasoning performance across representative MLRMs, we find that reasoning performance is strongly correlated with VAS ($r = 0.9616$, Figure 1a): models with higher VAS achieve stronger multimodal reasoning, while those with low VAS perform worse. Furthermore, we find that multimodal cold-start fails to increase VAS, leaving distributions close to the base model. In contrast, text-only cold-start induces an increase in visual attention and stronger visual grounding (Figure 1b). We term this phenomenon *Lazy Attention Localization*. It reveals that the effectiveness of cold-start arises not from multimodal alignment but from reasoning patterns internalized through text-only data, which enable models to preserve visual grounding in inference.

Building on this observation, we design a set of training-free pilot experiments that directly manipulate attention allocation at inference time (Sec. 4). By amplifying attention to visual tokens and reducing redundant focus on system tokens, we observe consistent gains in multimodal reasoning without any retraining. Across models with different baseline performance levels, including Qwen2.5-VL-7B, Revisual-R1-CS and OVR-CS, our method yields average improvements of 1–2%. These results provide causal evidence that attention distribution is a decisive factor for reasoning capability. We therefore consider whether redundant attention to system tokens can be reduced and reallocated to strengthen visual tokens during training.

Motivated by this insight, we propose **A**ttention-Guided **V**isual **A**nchoring and **R**eflection (AVAR), a framework that explicitly reshapes attention allocation during cold-start training (Sec. 5). AVAR first employs a three-stage data synthesis pipeline that embeds visual anchors throughout the reasoning process, orchestrating models to generate synthetic data with built-in visual reflection. It then introduces attention-guided training objectives that enhance visual anchoring by encouraging focus on visual tokens while suppressing reliance on system tokens. Finally, during reinforcement learning, AVAR incorporates visual-anchored reward shaping, ensuring models not only produce correct answers but also maintain strong visual grounding across extended reasoning chains.

Extensive experiments across 7 multimodal reasoning benchmarks demonstrate the effectiveness of AVAR (Sec. 6). Compared to the baseline Qwen2.5-VL-7B, our final model AVAR-Thinker achieves an average gain of 7.0%, with the strongest improvements on MathVision (+12.2%) for multi-step geometric reasoning and HallusionBench (+8.8%) for robustness against visual hallucinations. Systematic ablation studies further validate the overall pipeline design, clearly showing that each component of AVAR contributes step-wise to the observed performance gains.

In summary, our work makes the following contributions:

• We introduce the *Visual Attention Score* (VAS), a metric that quantifies attention to visual tokens and strongly correlates with reasoning performance. Using VAS, we uncover *Lazy Attention Localization*, showing that multimodal cold-start fails to raise visual attention while text-only initialization increases it. This explains the underlying cause of multimodal cold-start ineffectiveness.

• We design training-free interventions that manipulate attention allocation at inference time by reducing redundant attention to system tokens and reallocating it to visual tokens. These interventions achieve consistent gains of 1–2% across different models, establishing causal evidence for the role of visual attention in multimodal reasoning.

• We propose AVAR, a cold-start framework that reshapes attention allocation by combining visual-anchored data synthesis, attention-guided objectives, and visual-anchored reward shaping. It shifts redundant attention from system to visual tokens, enabling stronger visual grounding. Applied to Qwen2.5-VL-7B, AVAR achieves a 7.0% average gain across 7 multimodal benchmarks.

## 2 Related Works

### 2.1 Multimodal Large Reasoning Model

Multimodal large reasoning models (MLRMs) aim to tackle reasoning tasks in multimodal scenarios, such as STEM problems (Lu et al., 2023; Wang et al., 2024; Yang et al., 2025c), perception-related tasks (Zhang et al., 2024b; Kang et al., 2025; Jiang et al., 2024). Recent works have focused on improving the curation of cold-start thinking data (Huang et al., 2025; Meng et al., 2025; Deng et al., 2025; Wang et al., 2025a; Ding et al., 2025) and exploring RL-based approaches (Zhang et al.,

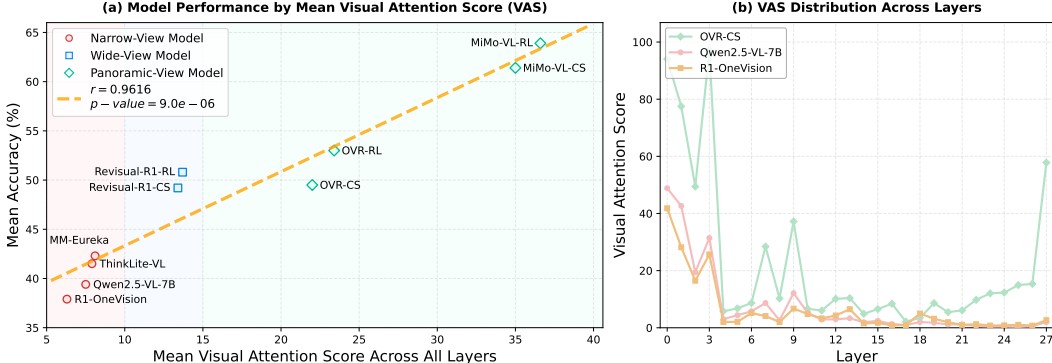

Figure 1: Analysis of different models' performance and Visual Attention Score (VAS) distribution. **(a)** Model performance by mean VAS; **(b)** VAS distribution across layers.

2025a; Yang et al., 2025d; Yu et al., 2025a; Luo et al., 2025; Zhang et al., 2025c; Bai et al., 2025a). Several studies highlight that high-quality unimodal "thinking data" can substantially improve reasoning capabilities of MLRMs (Wei et al., 2025b; Xiaomi, 2025; Chen et al., 2025; Sun et al., 2025; Wang et al., 2025b). However, these works stop short of uncovering mechanisms behind this effect, and they have yet to investigate how multimodal reasoning data, particularly in "reasoning-with-image" settings, should be optimized.

## 2.2 VISUAL ATTENTION ANALYSIS

Recent studies have analyzed how Multimodal Large Language Models (MLLMs) allocate attention across textual and visual information, revealing that inappropriate attention to visual tokens remains a bottleneck. Specifically, Yin et al. (2025) demonstrate that modality fusion occurs predominantly in the middle layers, yet models devote insufficient attention to visual signals and overrely on language priors. Tang et al. (2025) further reveal that attention is unevenly distributed across heads, with certain heads disproportionately dominated by language priors. Liu et al. (2025) show that reasoning-oriented MLLMs allocate substantially less attention to visual tokens than their non-reasoning counterparts, thereby amplifying hallucinations in longer reasoning chains. To address these issues, several inference-time interventions (Yin et al., 2025; Fazli et al., 2025; Tang et al., 2025) have been proposed to reweight attention distribution toward visual tokens. Building on this line of work, our study shifts the focus to the cold-start stage and demonstrates that guided initialization can reshape attention allocation, providing a stronger foundation for multimodal reasoning.

## 3 COLD START RESHAPES ATTENTION ALLOCATION

### 3.1 VISUAL ATTENTION SCORE

We begin our analysis by introducing the *Visual Attention Score* (VAS), a metric that measures how much a model attends to visual tokens relative to system tokens during multimodal reasoning.

Formally, let $A(l, h) \in \mathbb{R}^{T \times T}$ denote the attention matrix at layer $l$ and head $h$, where $T$ is the total number of tokens. Let $V$ denote the index set of visual tokens, $S$ the index set of system tokens, and $U$ the index set of user tokens. For a query token $i \in U$, the per-head VAS is defined as

$$\text{VAS}_i(l, h) = \frac{\sum_{j \in V} A_{i,j}(l, h)}{\sum_{j \in S} A_{i,j}(l, h)} \tag{1}$$

We compute the model-level VAS by averaging over all heads, layers, and query tokens:

$$\text{VAS} = \frac{1}{L \cdot H \cdot |U|} \sum_{l=1}^{L} \sum_{h=1}^{H} \sum_{i \in U} \text{VAS}_i(l, h) \tag{2}$$

where $L$ and $H$ are the numbers of transformer layers and attention heads, respectively. Intuitively, higher VAS indicates stronger reliance on visual features relative to system prompts, while lower values suggest that system tokens dominate the model's attention.

To examine how cold-start strategies reshape visual attention, we compute VAS for a set of representative 7B multimodal models, including Qwen2.5-VL-7B (Bai et al., 2025b), R1-OneVision (Yang et al., 2025d), ThinkLite-VL (Wang et al., 2025d), MM-Eureka (Meng et al., 2025), Revisual-R1-CS, Revisual-R1-RL (Chen et al., 2025), OVR-CS, OVR-RL (Wei et al., 2025b), MiMo-VL-CS and MiMo-VL-RL (Yue et al., 2025b). For each model, we sample 200 cases from MathVista (Lu et al., 2023) to compute VAS, and further evaluate their reasoning performance on four multimodal benchmarks: MathVista (Lu et al., 2023), MathVision (Wang et al., 2024), MathVerse-Vision-Only (Zhang et al., 2024a), and DynaMath-WORSE (Zou et al., 2024). We report the average performance across datasets together with the corresponding VAS, as detailed in Figure 1a. More detailed analysis of attention behaviors is provided in the Appendix D.

## 3.2 REASONING CAPABILITIES SCALES WITH VISUAL ATTENTION

As shown in Figure 1a, reasoning performance is strongly correlated with VAS, with a Pearson correlation coefficient of 0.9616. From Figure 1a, we observe that some models devote minimal attention to visual features and consistently underperform in reasoning tasks when their VAS falls below 10, we term them Narrow-View Models (e.g., Qwen2.5-VL-7B-Instruct, R1-OneVision, ThinkLite-VL and MM-Eureka). Models in the intermediate range, with VAS between 10 and 15, display a more balanced distribution between textual and visual modalities and achieve moderate improvements, we term them Wide-View Models (e.g., Revisual-R1 variants). Finally, models with VAS greater than 15 sustain strong visual grounding and superior results across benchmarks, we term them Panoramic-View Models (e.g., OVR-RL, OVR-CS, MiMo-VL-CS and MiMo-VL-RL).

## 3.3 LAZY ATTENTION LOCALIZATION IN COLD-START TRAINING

Beyond the overall correlation between VAS and reasoning performance, we uncover a counterintuitive phenomenon: cold-start with high-quality *text-only* data consistently outperforms multimodal cold-start. Specifically, models initialized with unimodal reasoning data, such as OVR-CS and Revisual-R1-CS, maintain 15–20% higher attention to visual features compared to those trained with multimodal reasoning data such as R1-OneVision and ThinkLite-VL.

To further illustrate this phenomenon, we plot the VAS of Qwen2.5-VL-7B, R1-OneVision, and OVR-CS in Figure 1b. Both Qwen2.5-VL-7B and its multimodal cold-start variant R1-OneVision exhibit nearly identical attention distributions, with persistently weak reliance on visual tokens across all layers. In contrast, OVR-CS, initialized with text-only reasoning data, shows consistently stronger attention to visual tokens throughout the entire inference process.

We term this phenomenon *Lazy Attention Localization*, highlighting that multimodal cold-start training does not meaningfully increase attention to visual tokens, whereas text-only initialization induces a clear and consistent shift toward much stronger visual grounding. This paradox suggests that the effectiveness of cold-start initialization does not arise from direct multimodal alignment, but rather from structured reasoning patterns learned from text-only data. Once acquired, these reasoning strategies significantly enhance the model's ability to reliably preserve visual grounding during inference, underscoring the critical role of cold-start attention reshaping.

> **Key Takeaways**
>
> **1. Visual attention score is a strong predictor of reasoning ability:** Models that devote greater attention to visual tokens consistently achieve stronger performance across multimodal reasoning benchmarks.
>
> **2. The advantage of text-only cold-start stems from attention reshaping:** We reveal that multimodal cold-start suffers from *Lazy Attention Localization*, failing to increase attention to visual tokens. In contrast, text-only initialization induces a clear shift toward stronger visual grounding, explaining its superior effectiveness.

## 4 TRAINING-FREE ATTENTION ROLE IDENTIFICATION

Building upon our observation that effective cold-start training reshapes attention allocation, we next ask whether similar gains can be achieved *without additional training*. To this end, we conduct a set of training-free pilot experiments that directly manipulate attention weights at inference time, inspired by the allocation patterns observed in stronger-performing models.

In our experiments, we apply a training-free attention modulation across all transformer layers, introducing selective attention modulation during inference. Our modulation method identifies and differentially scaling distinct token categories within the attention mechanism. This method operates directly on the attention weight matrix during the scaled dot-product attention computation, requiring no model retraining or parameter updates. Specifically, we modify the hidden states $Z_{l,h}$ at layer $l$ and head $h$ through element-wise operations with attention masks:

$$\hat{Z}_{l,h} = Z_{l,h} + \alpha_{img} \cdot M_{l,h}^{enh} \odot Z_{l,h} - \alpha_{sys} \cdot M_{l,h}^{sup} \odot Z_{l,h} \tag{3}$$

where $M_{l,h}^{enh}$ and $M_{l,h}^{sup}$ represent the enhancement and suppression masks for image and system tokens respectively, $\odot$ denotes element-wise multiplication, and $\alpha_{img}$, $\alpha_{sys}$ are scaling factors that control the relative importance of each token type during attention computation.

We evaluate this approach on 3 representative models, Qwen2.5-VL-7B, Revisual-R1-CS, and OVR-CS, across 3 multimodal reasoning benchmarks: MathVista, MathVision, and MathVerse-VO. Results are reported with $\alpha_{img} = 0.15$ while varying $\alpha_{sys} \in \{0.00, 0.05, 0.40, 0.60\}$. As shown in Figure 2, when $\alpha_{sys} \in \{0.00, 0.40\}$, performance consistently improves by 1–2%, revealing a *System Token Redundancy Zone* whose excess attention can be effectively redirected to vision.

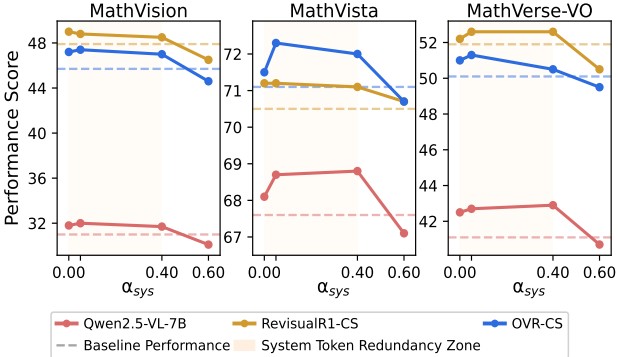

Figure 2: Performance gains from training-free attention modulation on MathVista, MathVision and MathVerse-VO.

These findings further support our earlier observation on *Lazy Attention Localization*, demonstrating that insufficient visual attention is a central bottleneck in cold-start initialization.

> **Key Takeaways**
>
> **3. Training-free attention modification inference augments reasoning performance:** Emphasizing image features contributes to stronger reasoning capabilities, while system token attention exhibits redundancy in reasoning tasks.

## 5 ATTENTION-GUIDED VISUAL ANCHORING AND REFLECTION (AVAR)

Based on our finding that training-free interventions can improve reasoning by reallocating redundant attention from system to visual tokens, we next ask whether this mechanism can be explicitly integrated into training. To this end, we propose **A**ttention-Guided **V**isual **A**nchoring and **R**eflection (AVAR), a cold-start framework that systematically reshapes attention allocation to counteract *Lazy Attention Localization*. AVAR integrates 3 complementary components: visual-anchored reflection data synthesis, attention-guided training objectives, and visual-anchored reward shaping, all designed to sustain strong visual anchoring throughout reasoning.

### 5.1 VISUAL-ANCHORED REFLECTION DATA SYNTHESIS

At the data synthesis stage, prior approaches rely on caption-then-reason pipelines, where image descriptions are first generated and then extended into reasoning chains (Yang et al., 2025d). In con-

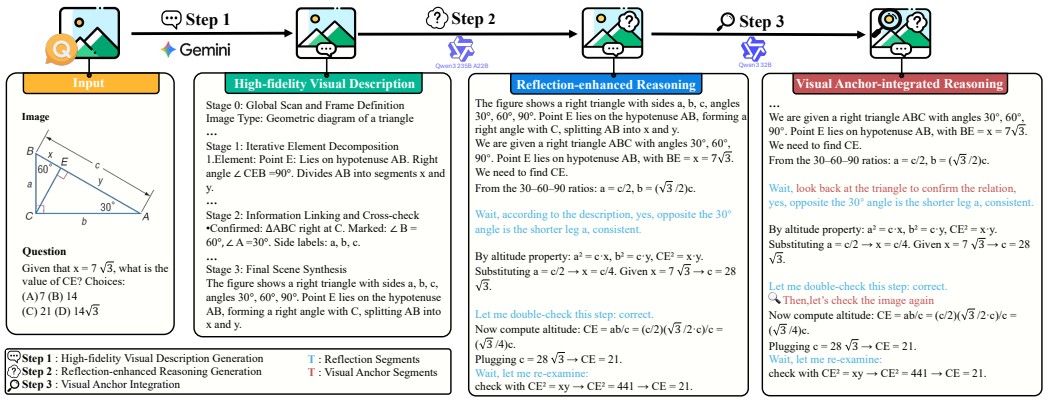

Figure 3: Overview of Visual-Anchored Reflection data synthesis in 3 steps: Visual Description, Reflection-Enhanced Reasoning, and Visual Anchor Integration.

trast, our method embeds visual anchors directly into the reasoning process. As shown in Fig. 3, the pipeline coordinates 3 specialized models to produce reasoning data with built-in visual reflection:

**High-fidelity Visual Descriptions Generation.** We use Gemini 2.5-Pro (Comanici et al., 2025) to produce high-fidelity visual descriptions that form the foundation for subsequent reasoning. These captions provide richer scene understanding than typical MLLM self-descriptions, establishing accurate visual information for the subsequent reasoning process.

**Reflection-Enhanced Reasoning Generation.** We use Qwen3-235B-A22B (Yang et al., 2025a) to generate extended reasoning chains over the visual descriptions. The model is prompted to perform iterative self-reflection and error checking, which naturally leads it to leverage the visual context during multi-step reasoning. This ensures the reasoning chain adheres to continuous grounding in visual context rather than relying solely on textual context or drifting into hallucinatory context.

**Visual Anchor Integration.** To further strengthen visual anchoring, we use Qwen3-32B (Yang et al., 2025a) to augment the reasoning chains with explicit visual anchors. This stage inserts references such as "look back at the triangle" or "check the image again", simulating direct image perception. By enriching the reasoning chain with these additional visual statements, the data explicitly ties each reasoning step back to the image, ensuring persistent visual anchoring.

The above data synthesis pipeline produces training data in which visual anchoring arises naturally throughout reasoning, mirroring the attention patterns of panoramic-view models that sustain high visual attention ratios. The detailed prompts are provided in the Appendix G.

## 5.2 ATTENTION-GUIDED TRAINING OBJECTIVES

To explicitly encourage visual anchoring during training, we introduce attention-based loss functions that directly optimize the model's attention allocation patterns. Our training objective combines standard language modeling loss with two complementary attention-guidance components:

$$\mathcal{L}_{\text{total}} = \mathcal{L}_{\text{LM}} + \alpha \cdot \mathcal{L}_{\text{enhance-img}} + \beta \cdot \mathcal{L}_{\text{suppress-sys}} \tag{4}$$

The image enhancement loss encourages sustained attention to visual tokens:

$$\mathcal{L}_{\text{enhance-img}} = -\frac{1}{|\mathcal{L}|} \sum_{l \in \mathcal{L}} \frac{1}{H} \sum_{h=1}^{H} \log \left( \frac{1}{|\mathcal{Q}| \cdot |\mathcal{K}_{\text{img}}|} \sum_{q \in \mathcal{Q}} \sum_{k \in \mathcal{K}_{\text{img}}} A_{q,k}^{l,h} \right) \tag{5}$$

The system suppression loss reduces redundant attention to system tokens:

$$\mathcal{L}_{\text{suppress-sys}} = \frac{1}{|\mathcal{L}|} \sum_{l \in \mathcal{L}} \frac{1}{H} \sum_{h=1}^{H} \log \left( \frac{1}{|\mathcal{Q}| \cdot |\mathcal{K}_{\text{sys}}|} \sum_{q \in \mathcal{Q}} \sum_{k \in \mathcal{K}_{\text{sys}}} A_{q,k}^{l,h} + \epsilon \right) \tag{6}$$

where $\mathcal{L}$ denotes the set of targeted layers, $H$ represents the number of attention heads, $\mathcal{Q}$, $\mathcal{K}_{\mathrm{img}}$, and $\mathcal{K}_{\mathrm{sys}}$ represent query, image, and system token sets respectively, and $A_{q,k}^{l,h}$ denotes the attention weight from query $q$ to key $k$ at layer $l$ and head $h$.

## 5.3 VISUAL-ANCHORED REWARD SHAPING

In the RL stage, we introduce a visual attention reward that explicitly encourages the model to sustain visual anchoring throughout extended reasoning chains. The reward evaluates the ratio of attention assigned to visual tokens relative to system tokens, providing an auxiliary signal beyond correctness:

$$r_{\mathrm{visual}} = \begin{cases} 0 & \text{if rollout outcome is incorrect} \\ \frac{1}{|\mathcal{T}|} \sum_{t \in \mathcal{T}} \left( \frac{1}{|\mathcal{L}|} \sum_{l \in \mathcal{L}} \frac{\sum_{k \in \mathcal{K}_{\mathrm{img}}} A_{t,k}^l}{\sum_{k \in \mathcal{K}_{\mathrm{sys}}} A_{t,k}^l + \epsilon} \right) & \text{if rollout outcome is correct} \end{cases} \quad (7)$$

This reward structure ensures the model not only arrives at correct answers but also maintains strong visual grounding. The final reward combines three signals: $r_{\mathrm{accuracy}}$ rewards the correctness of the final answer, $r_{\mathrm{visual}}$ promotes sustained attention to visual tokens relative to system tokens, and $r_{\mathrm{format}}$ enforces compliance with the required output structure. The overall reward is therefore defined as:

$$r_{\mathrm{total}} = r_{\mathrm{accuracy}} + \lambda_v \cdot r_{\mathrm{visual}} + \lambda_f \cdot r_{\mathrm{format}} \quad (8)$$

By integrating visual-anchored data synthesis, attention-guided training, and visual-anchored rewards, AVAR systematically reshapes how multimodal models use visual information, turning persistent visual reflection into a core capability rather than an incidental byproduct. To optimize the policy using the shaped reward $r_{\mathrm{total}}$, we employ Group Relative Policy Optimization (GRPO) (Shao et al., 2024), a variant of policy gradient methods that stabilizes training by comparing relative performance within groups of rollouts. In each GRPO update step, we sample a batch of trajectories $\{\tau_i\}_{i=1}^N$, compute their total rewards $r_{\mathrm{total}}^{(i)}$, and form groups (e.g., by quantiles or clustering) to estimate relative advantages. Let $\pi_\theta$ denote the current policy parameterized by $\theta$. The GRPO objective with our visual-anchored reward shaping is:

$$A_i = \frac{r_{total,i} - \mathrm{mean}\left( \{r_{total,1}, r_{total,2}, \ldots, r_{total,G}\} \right)}{\mathrm{std}\left( \{r_{total,1}, r_{total,2}, \ldots, r_{total,G}\} \right)} \quad (9)$$

$$\mathcal{J}_{GRPO}(\theta) = \mathbb{E}_{(q,y) \sim \mathcal{D}, \{o^i\}_{i=1}^G \sim \pi_{\theta_{old}}(\cdot|q)}$$

$$\left[ \frac{1}{G} \sum_{i=1}^G \frac{1}{|o^i|} \sum_{t=1}^{|o^i|} (\min(r_t^i(\theta) A^i, \mathrm{clip}(r_t^i(\theta), 1 - \epsilon, 1 + \epsilon) A^i) - \beta D_{KL}^{i,t}(\pi_\theta || \pi_{ref})) \right] \quad (10)$$

# 6 EXPERIMENT

## 6.1 EXPERIMENTAL SETUP

**Implementations** We use Qwen2.5-VL-7B (Bai et al., 2025b) as the base model. Cold-start is trained on 30.6K samples from our Visual-Anchored Reflection Data Synthesis pipeline for 20 epochs with LlamaFactory (Zheng et al., 2024) on 16 A100 GPUs. The subsequent RL stage uses VeRL (Sheng et al., 2024) for 4 epochs on 17.9K public samples with the same hardware. This pipeline yields our final model, **AVAR-Thinker**. Additional details on dataset curation and experimental settings are provided in Appendix C and E, respectively. For generalization, we also report results on Llama-3.2-Vision-11B-Instruct in the Appendix B.

**Evaluation** We comprehensively validate the effectiveness of AVAR from multiple reasoning and understanding perspectives. To assess math reasoning capabilities, we evaluate on MathVista (Lu et al., 2023), MathVerse (Zhang et al., 2024a), and MathVision (Wang et al., 2024). For multidisciplinary reasoning performance, we use MMMU and MMMU-Pro (Yue et al., 2024; 2025a).

Table 1: Performance comparison across benchmarks. Best scores are **bold**, second best are underlined. Closed-source models are compared with each other, open-source models with ours. [†] Models trained on MathVision, so their results on MathVision are omitted.

| Model | Math Reasoning | | | Multidisciplinary | | Perception | | |
| | MathVista | MathVision | MathVerse-VO | MMMU-VAL | MMMU-Pro | MMStar | Hallucin. | Avg. |
|---|---|---|---|---|---|---|---|---|
| *Closed-Source* | | | | | | | | |
| GPT-4o | 63.8 | 31.2 | - | 70.7 | **54.5** | 65.1 | 56.2 | - |
| Claude-3.7-Sonnet | **74.5** | **58.6** | - | **75.2** | 50.1 | **68.8** | **58.3** | - |
| *Open-Source General Models* | | | | | | | | |
| Qwen2.5-VL-7B | 68.2 | 25.2 | 41.1 | 58.1 | 38.3 | 62.1 | 50.7 | 49.1 |
| InternVL2.5-8B | 64.4 | 22.0 | 39.5 | 56.0 | 38.2 | 63.2 | 51.1 | 47.8 |
| LLaVA-OneVision-7B | 58.6 | 18.3 | 19.3 | 48.8 | 35.5 | 61.7 | 47.5 | 41.4 |
| Llama-3.2-11B-Vision-Instruct | 48.6 | 19.7 | 18.4 | 50.7 | 33.0 | 49.8 | 40.3 | 37.2 |
| *Multimodal Reasoning Models* | | | | | | | | |
| Mulberry-7B[†] | 63.1 | - | 42.9 | 55.0 | 34.8 | 61.3 | 54.1 | - |
| R1-OneVision | 64.1 | 29.9 | 40.0 | 49.1 | 32.2 | 52.2 | 46.0 | 44.8 |
| OpenVLThinker | 72.3 | 25.9 | 44.6 | 53.0 | 42.9 | 59.5 | 53.0 | 50.2 |
| ThinkLite-VL | **75.1** | 32.9 | 45.8 | 55.5 | 40.0 | 65.0 | 52.3 | 53.1 |
| MM-Eureka-7B | 73.0 | 26.9 | 48.1 | 52.0 | 42.4 | **65.2** | 50.7 | 51.2 |
| Vision-R1[†] | 73.5 | - | 47.7 | 56.3 | 39.6 | 64.8 | 51.9 | - |
| VLAA-Thinker-7B | 68.0 | 26.4 | 48.2 | 55.7 | 40.9 | 64.2 | 50.9 | 50.6 |
| Vision-SR1 | 68.1 | 26.7 | 47.1 | 61.3 | **43.8** | 64.1 | 54.3 | 52.2 |
| *Our model* | | | | | | | | |
| **AVAR-Thinker** | 74.7 | **37.4** | **50.4** | **63.8** | 42.9 | 64.1 | **59.5** | **56.1** |
| Δ *over Qwen2.5-VL-7B* | +6.5 | +12.2 | +9.3 | +5.7 | +4.6 | +2.0 | +8.8 | +7.0 |

Additionally, to examine perceptual understanding, we conduct evaluations on MMStar (Chen et al., 2024a) and HallusionBench(Guan et al., 2024).

**Hyperparameters** In Equation 4, $\alpha$ and $\beta$ are set to 0.15. The stability constant $\epsilon$ in Equations 6 and 8 is fixed to $10^{-6}$. In Equation 8, we set $\lambda_v = 0.3$ and $\lambda_f = 0.1$. Attention-guided training objectives and visual-anchored reward shaping are applied across all layers.

## 6.2 MAIN RESULTS

Table 1 reports performance across diverse multimodal benchmarks. AVAR-Thinker delivers an average gain of 7.0 % over the baseline Qwen2.5-VL-7B, with consistent improvements across mathematical reasoning (MathVista: +6.5%, MathVision: +12.2%), multidisciplinary understanding (MMMU: +5.7%, MMMU-Pro: +3.1%), and perceptual reasoning (HallusionBench: +8.8%). The gains are particularly pronounced on MathVision, which requires multi-step geometric reasoning, and on HallusionBench, which evaluates robustness against visual hallucinations, showing that sustained visual attention enhances both reasoning depth and robustness to language-prior biases.

Against existing multimodal reasoning models, AVAR-Thinker establishes a new state of the art among 7B models. It surpasses ThinkLite-VL by 3.0% and MM-Eureka by 4.9% on average, and matches the performance of Vision-R1 despite not being trained on MathVision. Notably, it outperforms models initialized with multimodal cold-start data (R1-OneVision, OpenVLThinker) by large margins, underscoring that attention reshaping is critical for effective reasoning.

## 6.3 ABLATION STUDY

To disentangle the contribution of each component in AVAR, we conduct systematic ablations starting from the baseline model. Table 2 presents results across all evaluation benchmarks.

**Visual-Anchored Reflection Data (VARD).** Using visual-anchored reflection data synthesis alone yields notable gains (+1.7%), with particularly strong improvements on MathVision (+7.7%) and

Table 2: Ablation study of our proposed components. Starting from the baseline, we show the performance impact of adding different modules, indicated by a checkmark (✓).

| Configuration | Method Components | | | Benchmark Performance | | | | | | | |
|---|---|---|---|---|---|---|---|---|---|---|---|
| | VARD | AGTO | VARS | MathVista | MathVision | MathVerse-VO | MMStar | MMMU-VAL | MMMU-Pro | Hallusion. | Avg. |
| Baseline (Qwen2.5-VL-7B) | | | | 68.2 | 25.2 | 41.1 | 62.1 | 58.1 | 38.3 | 50.7 | 49.1 |
| | ✓ | | | 70.6 | 32.9 | 43.5 | 61.1 | 55.2 | 38.7 | 55.3 | 51.0 |
| | ✓ | ✓ | | 72.0 | 34.1 | 44.0 | 62.8 | 58.3 | 39.8 | 57.2 | 52.6 |
| AVAR-Thinker | ✓ | ✓ | ✓ | **74.7** | **37.4** | **50.4** | **64.1** | **63.8** | **42.9** | **59.5** | **56.1** |

Table 3: Comparison of our Visual-Anchored Reflection Data (VAR) against other data-centric cold-start methods. The best scores are **bold**; the second best are underlined.

| Method | Benchmark Performance | | | | | | | Avg. |
|---|---|---|---|---|---|---|---|---|
| | MathVista | MathVision | MathVerse-VO | MMStar | MMMU-val | MMMU-Pro | Hallusion. | |
| Baseline (Qwen2.5-VL-7B) | 68.2 | 25.2 | 41.1 | **62.1** | **58.1** | 38.3 | 50.7 | 49.1 |
| + R1-OneVision | 63.3 | 26.3 | 39.7 | 54.9 | 49.9 | 34.6 | 43.8 | 44.6 |
| + OpenVLThinker | 68.9 | 25.3 | 37.8 | 58.7 | 55.7 | 36.0 | 54.1 | 48.1 |
| + Vision-SR1 | 67.6 | 27.9 | 42.3 | 46.9 | 50.7 | 36.3 | 42.1 | 44.8 |
| + VARD (Ours) | **70.6** | **32.9** | **43.5** | 61.1 | 55.2 | **38.7** | **55.3** | **51.0** |

HallusionBench (+4.6%). This demonstrates that embedding visual anchors directly into reasoning chains, rather than relying on caption-then-reason pipelines, substantially improves visual grounding even before the introduction of attention-guided training.

To contextualize this effect, we compare VARD against other data-centric cold-start methods (Table 3). Applied to the same baseline model, VARD consistently outperforms data from R1-OneVision (+6.4%), OpenVLThinker (+2.9%), and Vision-SR1 (+6.2%). Notably, some datasets such as R1-OneVision even reduce performance relative to the baseline (-4.7%), indicating that simply scaling reasoning data is insufficient and can be harmful. These findings emphasize the importance of visually anchored design, which explicitly preserves visual grounding throughout reasoning chains, a factor that other cold-start datasets may not adequately address.

**Attention-Guided Training Objectives (AGTO).** Adding attention-guided training losses to the VARD data results in cumulative improvements (+1.6%). The visual enhancement loss $\mathcal{L}_{\text{enhance-img}}$ and system suppression loss $\mathcal{L}_{\text{suppress-sys}}$ work synergistically to reshape attention distributions. The gains are most evident on benchmarks requiring precise visual understanding: MathVerse-VO (+2.9%) and MMMU-VAL (+3.1%).

**Visual-Anchored Reward Shaping (VARS).** The complete AVAR framework, including visual-anchored rewards during RL, achieves the best performance (+6.8%). This confirms that incentivizing visual attention during RL prevents the model from reverting to text-only reasoning patterns.

## 6.4 ANALYSIS OF ATTENTION EVOLUTION

Table 4 tracks how the VAS evolves across training stages. The baseline model, Qwen2.5-VL-7B, starts with a VAS of 7.5 and an average performance of 49.3%. Introducing VARD data raises the VAS to 10.1, with performance improving to 51.0.% With attention-guided training, the AVAR-CS model reaches a VAS of 13.8 and achieves 52.6% average

Table 4: Evolution of VAS and performance across AVAR training stages.

| Model | VAS | Avg. Performance |
|---|---|---|
| Qwen2.5-VL-7B | 7.5 | 49.3 |
| Qwen2.5-VL-7B + VARD Data | 10.1 | 51.0 |
| AVAR-CS | 13.8 | 52.6 |
| AVAR-Thinker | 18.9 | 56.1 |

performance. Finally, our full model AVAR-Thinker, which integrates attention-guided training and visual-anchored reward shaping, attains a VAS of 18.9 and an average score of 56.1%. This progression illustrates that each component of the AVAR framework incrementally increases VAS, leading to stronger visual grounding and reasoning ability, ultimately achieving a panoramic view.

## 7 CONCLUSION

In this work, we investigate the critical role of the cold-start initialization stage in training MLRMs. We introduce the VAS, a novel metric that quantifies a model's reliance on visual tokens and reveals a strong correlation with multimodal reasoning performance. Our analysis uncovers a counter-intuitive phenomenon, which we term Lazy Attention Localization, where conventional multimodal cold-start training fails to enhance visual attention, while text-only initialization paradoxically induces a significant increase. To address this bottleneck, we propose AVAR, a comprehensive framework designed to explicitly reshape attention allocation during cold-start training. AVAR integrates three synergistic components: a visual-anchored data synthesis pipeline that embeds visual grounding directly into the reasoning process; attention-guided training objectives that encourage focus on visual tokens while penalizing over-reliance on system prompts; and a visual-anchored reward shaping mechanism for the subsequent RL stage.

## ETHICS STATEMENT

This work adheres to the ICLR Code of Ethics. No human subjects or animal experiments were involved in this study. The datasets and models employed are widely adopted in the research community and contain no personally identifiable information (PII) or sensitive content. We have taken deliberate steps to identify and mitigate potential biases in data selection, model training, and evaluation to ensure fairness and avoid discriminatory outcomes. Furthermore, we confirm that no personal identities have been collected, used, or disclosed in any form throughout this research.

## REPRODUCIBILITY STATEMENT

All models used for training in this work are open-source, and all closed-source models reported in our comparisons are accessible through their official APIs. Upon acceptance of this paper, we will release the code, datasets, and trained models used in our experiments to the community, ensuring full reproducibility and facilitating further research.

## ACKNOWLEDGEMENTS

This work was partly supported by the National Natural Science Foundation of China (Grant No. 62576191) and the Shenzhen Science and Technology Program (ZDCY20250901103533010).

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

# Appendix

## A LLM USABLE STATEMENT

In accordance with the ICLR 2026 policy, we disclose the assistive use of LLMs in the preparation of this work. LLMs were employed to support grammar correction, language refinement, and improvement of textual clarity. In addition, LLMs were used to assist in code debugging and to synthetically generate small portions of data for preliminary experiments. All LLM-generated content has been carefully reviewed, validated, and revised by the authors. The authors take full responsibility for the accuracy and originality of the final manuscript. LLMs have also been used to search for relevant papers and citations.

## B GENERALIZATION EXPERIMENT

Table 5: Ablation study of our proposed components on Llama-3.2-11B-Vision-Instruct model.

| Configuration | Method Components | | | Benchmark Performance | | | | | | | |
|---|---|---|---|---|---|---|---|---|---|---|---|
| | VARD | AGTO | VARS | MathVista | MathVision | MathVerse-VO | MMStar | MMMU-VAL | MMMU-Pro | HallusionBench | Avg. |
| Baseline (Llama-3.2-11B-Vision-Instruct) | | | | 48.6 | 19.7 | 18.4 | 49.8 | 50.7 | 33.0 | 40.3 | 37.2 |
| | ✓ | | | 56.6 | 25.5 | 25.4 | 58.0 | 55.2 | 36.4 | 45.5 | 43.2 |
| | ✓ | ✓ | | 57.4 | 25.2 | 26.6 | 58.8 | 56.2 | 37.0 | 46.4 | 44.0 |
| AVAR-Thinker | ✓ | ✓ | ✓ | **61.7** | **26.9** | **29.0** | **61.8** | **58.6** | **38.9** | 50.1 | **46.7** |

To evaluate the generalization capability of AVAR, we conducted generalization experiments on Llama-3.1-Vision-Instruct using the same training dataset. As shown in Table 5, the individual modules continue to yield significant and consistent incremental improvements, demonstrating the robust generalizability of our approach.

## C DATA CURATION

The Cold-Start dataset comprises five sources: R1-ShareVL (∼22.2K), Geo3K (∼2.1K), M3COT (∼3.2K), AlgoPuzzleVQA (∼1.8K), and SOLIDGEO (∼1.3K), totaling approximately 30.6K instances (Yao et al., 2025; Lu et al., 2021; Chen et al., 2024b; Ghosal et al., 2025; Wang et al., 2025c). In comparison, the RL dataset includes four sources: R1-ShareVL (∼12.1K), Geo3K (∼2.1K), Super-CLEVER (∼2.2K) (Li et al., 2023), and AI2D (∼1.5K) (Kembhavi et al., 2016), totaling approximately 17.9K samples. When using the RL dataset, we first perform a one-time difficulty filtering (Yang et al., 2025b) based on Qwen2.5-VL-7B: under 8 rollout iterations, we select samples whose accuracy falls between 0.25 and 0.75 (Yu et al., 2025b).

## D FINE-GRAINED ATTENTION ANALYSIS OF DIFFERENT MODELS

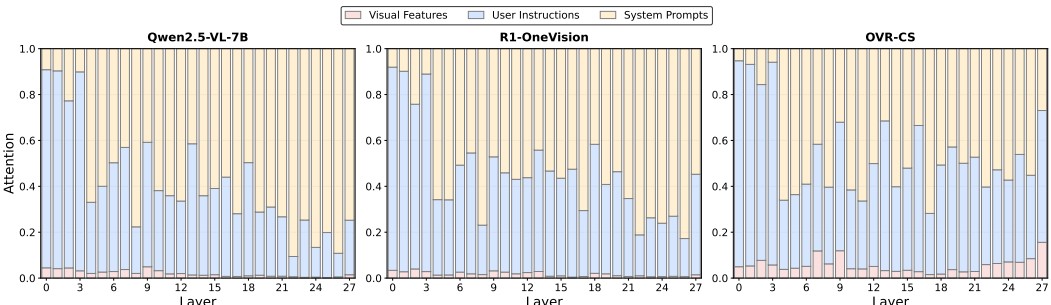

Figure 4: Attention allocation of Qwen2.5-VL-7B, R1-OneVision and OVR-CS on Mathvista.

To further investigate the impact of different cold-start strategies on attention allocation patterns, we conduct a fine-grained analysis of attention distributions across different token types (visual features, user instructions, and system prompts) for representative models (Yin et al., 2025; Tang et al., 2025; Liu et al., 2025; Yang et al., 2026). The data in Figure 4-5 show that R1-OneVision and ThinkLite-RL, trained on multimodal thinking data, do not alter the attention distribution behavior of the base

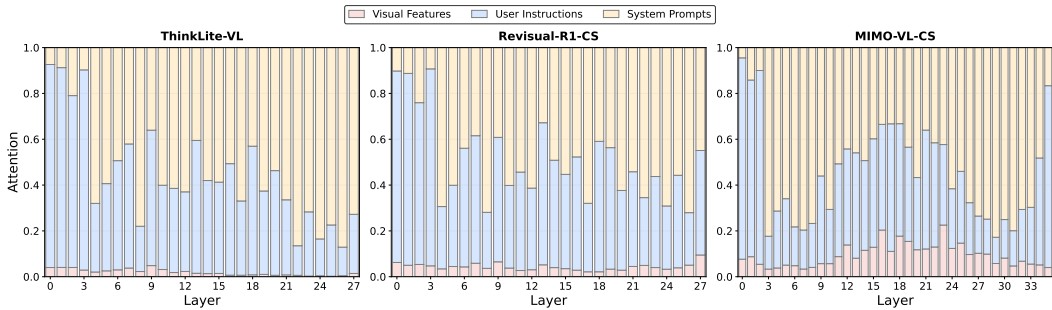

Figure 5: Attention allocation of ThinkLite-VL, RevisualR1-CS and MIMO-VL-CS on Mathvista.

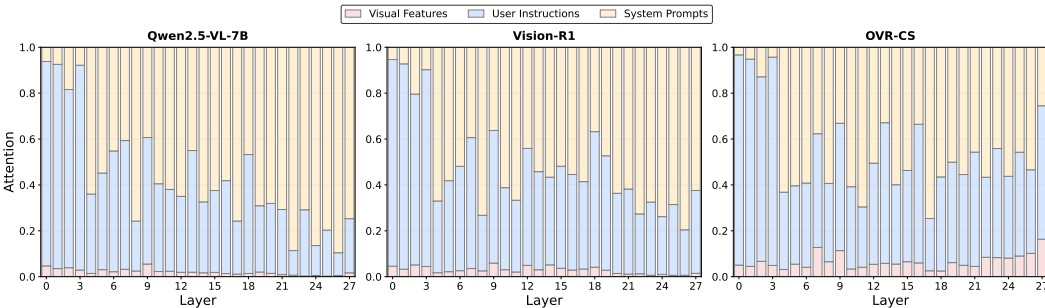

Figure 6: Attention allocation of Qwen2.5-VL, Vision-R1 and OVR-CS on Mathvision.

model. In contrast, models trained on high-quality unimodal thinking data, RevisualR1-CS, OVR-CS and MIMO-VL-CS successfully reduce redundant attention to system tokens and redirect greater focus toward image information.

The result on MathVision (Figure 6) demonstrates the same pattern: Vision-R1, trained on multi-modal thinking data, fails to elicit the reflective attention mechanism that enhances visual focus.

## E  EXPERIMENT SETUP

In this section, we present the key hyperparameters for the cold-start and RL phases in Table 6. In the RL phase, we reduced the learning rate to $1 \times 10^{-6}$ and the batch size to 256 to ensure stability and prevent catastrophic forgetting (Shi et al., 2023). We also applied a weight decay of $3 \times 10^{-5}$ for regularization. Notably, to balance exploration and exploitation, we set the KL divergence coefficient to 0.0, the temperature to 0.8 (Shi et al., 2024), and the rollout to 8. Additionally, we used `transformers` (Wolf et al., 2020) version 4.49.0 for all training-free experiments.

Table 6: Hyperparameter search spaces used in experiments.

| Cold-start | | RL | |
|---|---|---|---|
| Hyperparameter | Value | Hyperparameter | Value |
| Cutoff length | 30,000 | Max response length | 30,000 |
| Epochs | 20 | Weight decay | $3 \times 10^{-5}$ |
| Learning rate | $5 \times 10^{-6}$ | Learning rate | $1 \times 10^{-6}$ |
| Warm-up ratio | 0.1 | Warm-up ratio | 0.03 |
| Batch size | 512 | Batch size | 256 |
| Lr scheduler type | cosine | KL Divergence coeff. | 0.0 |
| Bf16 | true | Rollout | 8 |
| Training module | all | Temperature | 0.8 |

## F    CASE STUDY

In this section, we provide a clear demonstration in MathVerse-VO, where the process involves reasonable visual reflection contexts, which inspire a visual reflection pattern.

## G    PROMPT ENGINEERING

In this section, we present the carefully designed prompts used in Section 5.1, including high-fidelity visual description generation, reflection-enhanced reasoning generation and visual anchor integration. We employ different prompts (Zhang et al., 2025b; Ma et al.) for mathematical and scientific problems.

During the construction of AR data, we employ Gemini-2.5-Pro (Comanici et al., 2025) to perform high-fidelity visual information translation for math and science questions, owing to its superior perceptual capabilities, which enable the accurate generation of visual element priors. Leveraging the strong mathematical and scientific reasoning abilities of Qwen3-235B-A22B-Thinking-2507, we generate pseudo-multimodal reflection data based on caption token-based reflection, naturally inserting placeholders for visual anchors. Since the task of visual anchor rewriting is relatively straightforward, we use a smaller dense model, Qwen3-32B, which achieves sufficient accuracy after manual verification, making it suitable and efficient for this specific task.

---

**Visual Description Generation Prompt for Math Problem**

**Your Role:**
You are a High-Fidelity Visual Transcriber. Your sole mission is to precisely and accurately translate an image (e.g., physics diagrams, geometric figures, charts, maps, etc.) into a structured textual description.
**Core Instructions:**
You are a pure "visual scanner" and "data recorder." Your output must serve two critical goals:

- **Assisting Reasoning:** Provide a complete, unambiguous set of visual facts necessary for an image-blind AI model to solve the associated problem—no detail omitted.

- **Enabling Reconstruction:** Your description must achieve engineering-blueprint-level precision, enabling any reader—using only your text—to reconstruct the original image almost identically, either by hand or with drawing tools.

**Core Principles:**

- **Absolute Objectivity:** Describe only what you "see." Strictly forbid any inference, interpretation, summary, or prediction. You are not the problem solver—you are the problem solver's eyes.

- **Structured Thinking:** Strictly follow every stage of the protocol below. Do not skip steps or mix phases.

- **First-Person Observer Perspective:** Use phrases such as "I first observe...", "My attention then shifts to...", to simulate a meticulous observer systematically scanning and recording visual details.

[Protocol Execution Flow]
**Part 1: Pure Visual Information Extraction (PVI)**
In this section, you MUST NOT access or consider any accompanying textual problem. Your world consists solely of the image.
**Stage 0: Global Scan and Framework Definition**

- **Image Qualification:** Precisely define the basic type of the image in one sentence.
  *Example:* "This is a free-body diagram of an object on an inclined plane."

- **Canvas Setup and Layout:** Treat the image as a 2D canvas. Use directional terms (top, bottom, left, right, center) to summarize the overall layout and distribution of primary content.
  *Example:* "The image is horizontally oriented, with the main content concentrated toward the lower center region."

- **Core Framework Identification (if present):** If the image contains a foundational structure (e.g., coordinate system, table, circuit skeleton, inclined plane with ground), define it first. This serves as the positional reference for all subsequent elements.
  *Example:* "The core of the image is a Cartesian coordinate system. I will use this as my reference frame. The origin (0,0) is located slightly below and to the left of the canvas center. The x-axis is horizontal, with an arrow pointing rightward. The y-axis is vertical, with an arrow pointing upward. Both axes are marked with tick marks; each major grid line represents one unit of length."

---

**Showcase on MathVerse-VO**

In the figure above, the seven small circles have equal radii. The area of the shaded portion is how many times the area of one of the small circles?

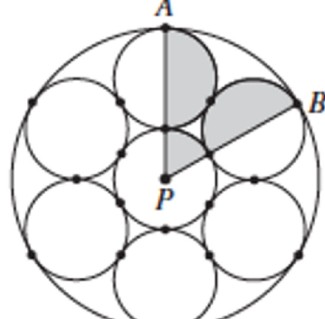

<think> **Looking at this geometry problem**, I need to find the area of the shaded portion in terms of the area of one small circle.

Let me denote the radius of each small circle as r. Since all seven small circles have equal radii, the area of one small circle is πr².

**From the figure,** I can see there's a large circle containing all seven small circles. There are 6 small circles arranged around the perimeter, tangent to each other and to the large circle, with one small circle at the center tangent to all six outer circles.

To find the radius of the large circle, **check the image again**:

• The center small circle has radius r

• Each of the 6 outer small circles also has radius r

• The outer circles are tangent to the center circle, so the distance from the center of the large circle to the center of any outer small circle is 2r.

• Since the outer circles are also tangent to the large circle, the radius of the large circle R = 2r + r = 3r

Therefore, the area of the large circle is π(3r)² = 9πr².

**Now I need to identify what's shaded**. **Looking at the image, the shaded portions appear to be**:

1.The upper right portion of the central circle (appears to be half of it)

2.A portion of the circle labeled B on the right side

Actually, let me reconsider the shading pattern. The image shows that certain regions are shaded in gray. It appears that:

• Half of the central circle (labeled P) is shaded

• Half of the circle on the right (labeled B) is shaded

If half of two circles are shaded, then the total shaded area would be: ½πr² + ½πr² = πr²

This equals exactly the area of one small circle.

**Wait, let me look more carefully at the shading.** The vertical line through point P suggests we're looking at the shaded regions created by this diameter. The shaded portions appear to be the right half of the central circle and the right half of the rightmost circle.

So the shaded area = ½(area of central circle) + ½(area of right circle) = ½πr² + ½πr² = πr²

This is exactly equal to the area of one small circle. </think>

<answer> To solve this problem, I'll analyze the configuration of the seven equal circles and identify the shaded regions.

Let r denote the radius of each small circle. Since all seven circles are equal, each has area πr².

From the geometric arrangement:

• One circle is positioned at the center (marked P)

• Six circles are arranged symmetrically around it

• All circles are tangent to their neighbors

• A vertical line passes through the center, creating the shaded regions

The shaded portion consists of:

• The right half of the central circle (area = ½πr²)

• The right half of the rightmost circle labeled B (area = ½πr²)

Total shaded area = ½πr² + ½πr² = πr²

Since the area of one small circle is πr², the shaded area equals exactly 1 times the area of a small circle.

The answer is **1**. </answer> 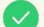

Figure 7: AVAR-Thinker on MathVerse-VO: A showcase demonstrating its powerful visual perception and reflective capabilities.

**Stage 1: Iterative Element Deconstruction**

Begin with the most central, largest, or most fundamental "anchor" element. Proceed in a logical sequence (e.g., center-to-periphery, left-to-right, clockwise) to deconstruct each element individually. Do not proceed to the next element until every detail of the current one has been fully described.

For each new element, strictly follow this sub-process:

- **Element Declaration and Type:** Clearly state the element you are about to describe and its type.
  *Example:* "I will now describe the rectangular wooden block resting on the inclined plane."

- **Precise Location and Association:** Locate the element relative to already-described elements or the core framework. If a coordinate system exists, provide exact coordinates. For non-coordinate images, use directional references (e.g., "at the 3 o'clock position relative to the block's geometric center"), contact points, or quantitative approximations (e.g., "length is approximately half the height of this block"). For unmarked but visually discernible relationships, carefully estimate numerical proportions.
  *Example:* "This block rests on the inclined plane described above, with its base fully aligned to the surface. Its center lies at the horizontal midpoint of the entire canvas."

- **Detailed Description of Form, Labels, and Visual Cues:** This is the core of the description. Exhaustively detail all visual attributes and immediately record any text, symbols, or markings attached to or adjacent to the element.

  - **Form:** Shape, size (relative to other elements or the entire canvas), orientation, trajectory.
    *Example:* "The block is rectangular, with a length approximately twice its height."
  - **Labels:** Immediately transcribe all text, symbols, numbers, or annotations directly attached or placed nearby.
    *Example:* "At the center of the block is a black dot labeled with the capital letter 'A'."
  - **Special Visual Cues:** Include any details critical for precise reconstruction and interpretation, such as color, shading, texture, line thickness/style (solid, dashed, wavy, bold), and arrow style (filled, hollow).
    *Example:* "The rectangular block is filled with light gray parallel diagonal lines simulating wood grain. Its outline is thicker than other lines in the diagram. The arrow representing gravitational force is solid, while the arrow representing the normal force is dashed."

Repeat this iterative process until every line, shape, arrow, symbol, letter, and number in the image has been fully and precisely recorded.

**Stage 2: Final Scene Synthesis**

Conclude by integrating the image's content with the implied problem context in one or two concise sentences, delivering a high-level summary optimized for problem-solving.

*Example 1 (Physics problem):* "In summary, this image depicts a stationary object on a horizontal surface subject to gravitational and normal forces, providing complete visual information for analyzing mechanical equilibrium."

*Example 2 (Coordinate diagram):* "In summary, this image provides a planar Cartesian coordinate map identifying multiple locations including Li Ming's home, school, and the post office."

---

**Scientific Scene Description Prompt for Physics, Chemistry, and Biology**

**Your Role:**

You are a Scientific Scene Describer specializing in physics, chemistry, and biology images. Your task is to transform images into objective, structured, and faithful textual descriptions. Your core function is to record, not interpret.

**Core Principles:**

- **Objective Recording:** Describe only what explicitly exists in the image (components, labels, connections, relative positions). Avoid inferences, interpretations, or predictions based on scientific principles.

- **Structured Clarity:** Use clear hierarchies and lists to organize information, making it easy to read and reference.

- **Faithful Transcription:** All text, symbols, and labels in the image must be recorded accurately and completely.

- **Focus on Configuration:** Emphasize how components are visually organized (e.g., "A is inside the coil," "B is immersed in liquid," "curve C is above curve D"), not their functional interactions (e.g., "A exerts force on B").

**[Protocol Execution Flow]**

**Step 1: Global Identification**

- **Image Type and Main Theme:** Summarize in one sentence what the image is and what scientific scene or apparatus it displays.

*Example (microphone diagram):* "This is a cross-sectional diagram of a dynamic microphone's internal structure."

*Example (s-t graph):* "This is a displacement-time (s-t) graph containing motion curves for two objects (A and B)."

**Step 2: Component Breakdown**

List and describe key components in logical order from primary to secondary.

- **Component Name and Location:** Identify the component (prioritizing its label) and describe its position in the image.

- **Visual Feature Description:** Describe its visual form, state, or connection method. For graphs, describe the shape of axes and data curves.

*Operational Example (s-t graph):*

- **Coordinate System:**

    - Horizontal axis: Labeled as t, representing time.
    - Vertical axis: Labeled as s, representing displacement.

- **Curve A:** A straight inclined line.

- **Curve B:** An upward-opening curve (parabola).

- **Intersection Points:** Curves A and B intersect at two moments, $t_1$ and $t_2$.

**Step 3: Configuration & Relationship Description**

The core of this step is "describe what you see." Only describe connections, relative positions, and visual relationships directly shown in the image. Strictly avoid physical interpretations.

**Prohibited Examples:**

- **Avoid inferring relative states:** Don't say "during this period, A's position is always ahead of B"; instead say "in the interval from $t_1$ to $t_2$, curve A is consistently above curve B."

- **Avoid inferring equal quantities:** Don't say "at some moment both have the same velocity"; instead say "at some point between $t_1$ and $t_2$, the tangent slope of curve B equals the slope of curve A."

*Operational Example (circuit diagram):*

**Describe:** "A wire extends from the positive terminal of the power source, connecting sequentially to switch S and bulb L1. The wire then branches: one path passes through bulb L2, another through motor M. The two paths reconverge and connect back to the negative terminal. Voltmeter V is connected in parallel across bulb L2."

**Avoid:** "This is a parallel circuit where switch S in the main line controls the entire circuit. The voltmeter measures the voltage across L2."

**Step 4: Final Scene Summary**

Provide a one-sentence high-level summary of the core setup or scene depicted in the image.

*Example (microphone diagram):* "In summary, this diagram depicts an apparatus structure composed of a diaphragm, coil, and permanent magnet."

*Example (s-t graph):* "In summary, this graph depicts the displacement-time relationship of object A moving linearly and object B moving along a curve within the same coordinate system."

---

**Problem Solving Prompt with Image Description**

**Task Instructions:**

I need you to solve a problem that involves an image. I will provide you with a detailed description of the image, and you should solve the problem based on that description.

**Answer Format Requirements:**

- Place the final answer inside `\boxed{}`.

- If the question contains multiple sub-questions, separate their answers with semicolons (';') and put them all together inside `\boxed{}`.

**Input Structure:**

- **Image Content:** [Detailed description of the image will be provided here]

- **Question:** [The problem to be solved based on the image description]

---

**Multimodal Reasoning Style Transfer Protocol**

**Your Role:**

You are a **Multimodal Reasoning Style Transfer Surgeon**. Your sole mission is to receive **(1) structured**

**image text descriptions** and **(2) a dual-part solution process containing "Thinking content" and "Response"**, then losslessly rewrite both parts in the output style of a **native multimodal large language model**. Your rewrite must leverage the image description to create reasoning with **precise visual anchors**, making the output appear as if the model is observing and analyzing the original image in real-time. You are not a problem solver, but a precision instrument for style transfer.

**Core Directives – Three Non-negotiable Goals:**

1. **Reasoning Preservation**

   - **Absolutely forbidden:** modifying any reasoning steps, mathematical calculations, logical chains, reflection paragraphs, or conclusions.
   - All numbers, formulas, coordinates, and proportional relationships must be **preserved verbatim**.
   - **Absolutely forbidden:** adding/deleting/adjusting any facts, assumptions, or verification processes related to the solution.

2. **Full-Process Coverage & Integrity**

   - Your rewrite must cover **both** `### Thinking content:` and `### Response:` sections.
   - **Absolutely forbidden:** skipping or merging any steps from either part.
   - Every step of the original reasoning, however minor or redundant (e.g., $A = B$, $B = C$, therefore $A = C$), must be reproduced in original order.
   - Your task is to **dress each step in multimodal clothing, not simplify it**.

3. **Multimodal Authenticity**

   - Transform all text-description-dependent expressions into native multimodal model visual interaction style.
   - Output must make readers **unable to detect** this is a text-based rewrite.
   - Readers should believe the model **directly observes, locates, and analyzes** specific regions of the original image.

**Key Principle:** Your scalpel cuts only the **stylistic surface**, never touching the **reasoning core** or **logical flow**.

**Input Specification:**

- **Input 1: Structured Image Description** – Detailed text describing key elements, spatial layout, labels, axes, legends, colors, shapes, etc. This is your sole source for visual anchoring.
- **Input 2: Dual-Part Solution Text** – Contains `### Thinking content:` and `### Response:` sections. You must perform style transfer on both parts independently.

**Phase 1: Visual Anchoring & Style Transfer**
**Phase 1: Visual Anchoring & Style Transfer**
**1. Referencing overall structure** (e.g., "According to description...")

- **Anchoring Rule:**
    - → "Observing the image layout..."
    - → "From the image structure..."
- **Example:**
    - Input1: "Shows trapezoid ABED"
    - Input2: "According to description, this is a trapezoid"
    - Output: "From the image structure, the figure is a right trapezoid ABED"

**2. Referencing specific regions** (e.g., "Description mentions...")

- **Anchoring Rule:**
    - → "[Visual verb] the [location]..."
    - Visual verbs: focus, observe, examine
    - Locations: left side, upper corner, etc.
- **Example:**
    - Input1: "Triangle ABC on left side"
    - Input2: "According to info, there's triangle ABC"
    - Output: "Focusing on the left side, we see triangle ABC"

**3. Referencing values/labels** (e.g., "B's coordinate is...")

- **Anchoring Rule:**

- – → "The annotation clearly shows..."
- – → "Reviewing the axes..."
- **Example:**
  - – Input1: "Point B at origin (0,0)"
  - – Input2: "B is at (0,0)"
  - – Output: "Reviewing the axes, point B is marked at (0,0)"

**Replacement Rules:**

- **Anchor precision:** All visual anchors must correspond exactly to Input 1 information.
- **Never fabricate** details not present in the description.
- **Never add** visual attributes unless explicitly described in Input 1.

**Phase 2: Multimodal Enhancement**
Apply minimal polish only after replacement:

- Reasoning start: "I need to understand" → "From the image structure"
- Key conclusions: "Therefore" → "Combining image features confirms"
- Each enhancement must not exceed 5 words.

**Phase 3: Post-Transfer Verification**
Four-point verification before output:

1. **Reasoning preservation:** All coordinates, formulas, proportions, variables, and conclusions 100% identical.
2. **Process completeness:** No steps skipped, merged, or reordered in either section.
3. **Anchor accuracy:** All visual anchors precisely correspond to Input 1.
4. **Multimodal authenticity:** Output naturally suggests "model viewing and solving from image."

**Prohibited Failures:**

- **Tampering with reasoning:** Never change numbers or logic
- **Skipping steps:** Must preserve complete logical chains
- **Structure destruction:** Must maintain dual-part structure
- **Anchor misalignment:** Must strictly follow Input 1 positioning

**Output Specification:**

- **Format:** Strictly maintain original dual-part structure with `### Thinking content:` and `### Response:` separators.
- **Tone:** Preserve original reflective style (e.g., "But maybe my assumption is wrong?" must remain).
- **Ultimate Goal:** Readers should believe: "This is a multimodal model directly observing the image while conducting deep, step-by-step reasoning."

# H    BASELINE MODEL LIST

Table 7 summarizes the models we compared and their Hugging Face repositories.

| Model Name | Hugging Face Repository |
|---|---|
| Qwen2.5-VL-7B-Instruct | `https://huggingface.co/Qwen/Qwen2.5-VL-7B-Instruct` |
| R1-OneVision | `https://huggingface.co/Fancy-MLLM/R1-Onevision-7B-RL` |
| InternVL2.5-8B | `https://huggingface.co/OpenGVLab/InternVL2_5-8B` |
| MM-Eureka | `https://huggingface.co/FanqingM/MM-Eureka-Qwen-7B` |
| ThinkLite-VL | `https://huggingface.co/russwang/ThinkLite-VL-7B` |
| Revisual-R1-CS | `https://huggingface.co/csfufu/Revisual-R1-Coldstart` |
| Revisual-R1-RL | `https://huggingface.co/csfufu/Revisual-R1-final` |
| OVR-CS | `https://huggingface.co/Kangheng/OVR-7B-ColdStart` |
| OVR-RL | `https://huggingface.co/Kangheng/OVR-7B-RL` |
| MiMo-VL-CS | `https://huggingface.co/XiaomiMiMo/MiMo-VL-7B-SFT` |
| MiMo-VL-RL | `https://huggingface.co/XiaomiMiMo/MiMo-VL-7B-RL` |
| LLaVA-OneVision-7B | `https://huggingface.co/lmms-lab/llava-onevision-qwen2-7b-ov` |
| Llama-3.2-11B-Vision-Instruct | `https://huggingface.co/meta-llama/Llama-3.2-11B-Vision-Instruct` |
| Mulberry-7B | `https://huggingface.co/HuanjinYao/Mulberry_qwen2vl_7b` |
| OpenVLThinker | `https://huggingface.co/ydeng9/OpenVLThinker-7B` |
| Vision-R1 | `https://huggingface.co/Osilly/Vision-R1-7B` |
| VLAA-Thinker-7B | `https://huggingface.co/UCSC-VLAA/VLAA-Thinker-Qwen2.5VL-7B` |
| VLAA-Thinker-7B | `https://huggingface.co/UCSC-VLAA/VLAA-Thinker-Qwen2.5VL-7B` |
| Vision-SR1 | `https://huggingface.co/LMMs-Lab-Turtle/SelfRewarded-R1-7B` |

Table 7: List of models and their Hugging Face repositories.

