# OpenReview forum: "From Narrow to Panoramic Vision: Attention-Guided Cold-Start Reshapes Multimodal Reasoning"
_ICLR.cc/2026/Conference — ICLR 2026 Poster_

### Official Review · Reviewer_LShU · 2025-10-25

**Soundness:** 2
**Presentation:** 3
**Contribution:** 2
**Rating:** 4
**Confidence:** 3

**Summary:**

The paper focuses on the mechanisms and effects of multimodal large reasoning models (MLRM) during the cold-start stage, proposes the Visual Attention Score (VAS) to quantify “visual token attention,” and empirically shows a strong correlation between average VAS and multimodal reasoning performance (the paper reports Pearson r=0.9616). The authors further observe that adopting a multimodal cold start does not improve VAS, whereas a pure text cold start significantly increases VAS; this counterintuitive result is termed Lazy Attention Localization (LAL).
Based on this diagnosis, the authors first conduct inference-time (training-free) attention redistribution experiments. By amplifying attention to visual tokens and suppressing attention to “system” tokens, they achieve a consistent gain of about 1–2% across multiple benchmarks and propose the existence of a “System Token Redundancy Zone” that can be reallocated to visual attention.
The authors then propose a three-part cold-start framework, AVAR: (1) Visual-Anchored Reflection Data synthesis (VARD), which embeds explicit anchors such as “look back at the image/check the figure” in the chain of thought; (2) Attention-Guided Training Objective (AGTO), which uses \mathcal{L}{\text{enhance-img}} and \mathcal{L}{\text{suppress-sys}} to encourage attention to vision and suppress attention to system tokens, respectively; (3) Visual-Anchored Reward Shaping (VARS), which uses an attention ratio as an auxiliary reward during RL and combines it with accuracy and formatting rewards to form the total return.
Using Qwen2.5-VL-7B as the base, the authors report an average improvement of +7.0% on seven multimodal reasoning benchmarks. Ablations indicate “stepwise” contributions from the three components, and VAS increases progressively with training stages (7.5→10.1→13.8→18.9).

**Strengths:**

1. Treats “visual attention allocation” as the core explanatory variable for cold-start effectiveness, proposes VAS and systematically links it to reasoning performance, introduces and names the phenomenon Lazy Attention Localization, and provides a coherent diagnosis–intervention–training framework (VAS→inference-time intervention→training-time objective/reward), with a complete rationale and a clear trajectory.

2. Reports results on multiple public benchmarks (MathVista/MathVision/MathVerse-VO, MMMU/Pro, MMStar, HallusionBench), compares against various 7B-scale multimodal reasoning models, presents overall and per-benchmark improvements, and further details ablations and stage-wise VAS changes, forming a relatively solid chain of evidence.

3. Provides a formal definition of VAS (taking the ratio of attention from user query tokens to visual/system keys as the unit, then averaging across layers, heads, and tokens) and clear mathematical expressions of the losses/rewards, facilitating implementation and reproducibility.

4. If the observed LAL is widespread, attention reshaping for cold starts could become a general recipe for multimodal reasoning models, with potential value for both academic and industrial training paradigms.

**Weaknesses:**

1. Although a strong correlation of r=0.9616 is reported, there is a lack of stricter causal testing for VAS→performance and control of confounders (e.g., differences in visual encoders across models, differences in system prompt templates, decoding length/temperature, etc.). At present, “inference-time attention amplification” yields only a 1–2% gain, which makes it difficult to explain the entire 7% average improvement during training. Suggested additions: controlled experiments that remove/replace the visual encoder, fix the system prompt template, and test sensitivity to decoding length/temperature.

2. Using “attention to system tokens” as the denominator of VAS requires justification: the number and positional distribution of system tokens can vary widely across data/templates, potentially introducing bias. There is a lack of comparisons against alternative metrics that use user text tokens or the number of image patches as references and of consistency results. In addition, no sensitivity analyses at the layer/head/position levels are presented in the main text to support the robustness of the main conclusions (Appendix D is mentioned, but key statistics are not shown in the main body).

3. Although reproducibility is claimed, the VARD stage uses closed-source models such as Gemini 2.5-Pro to generate high-fidelity descriptions, which may make it difficult for the community to replicate the same data quality. Meanwhile, the prompts, filtering, and quality-control details of the three-model collaborative pipeline are said to be in Appendix G, but the main text does not provide enough “actionable recipe” details (e.g., sampling temperature, rejection strategies, QC thresholds). On the implementation side, only coarse-grained parameters such as 30.6K / 20 epochs are given, with no disclosure of random seeds and variance.

4. The main improvements are concentrated in visual mathematics scenarios such as MathVision/MathVerse-VO. Although MMStar/MMMU are included, there is a lack of dimensions such as real-world high-resolution VQA, video temporality, and multi-image multi-turn reasoning, making it difficult to judge the robustness and cost (inference latency/memory) of the method across a broader family of tasks.

5. There is no systematic comparison against classic training-time approaches such as attention regularization/pruning/head reweighting or visual token resampling/token-dropping. The inference-time intervention is also not compared in parallel, under equal compute budget and equal parameter changes, with existing “adaptive attention calibration/head-level intervention” methods.

6. Tables 1/2/4 mostly report point estimates without confidence intervals/significance tests, and they do not report mean±variance over multiple random seeds, making it difficult to assess the robustness of the +1–2% and +3–7% gains.

7. Suppressing attention to system tokens may weaken instruction following and safety (many safety/formatting control signals depend on system prompts), yet the paper focuses only on performance improvements and does not provide comparative evaluations or failure analyses for instruction compliance and safe refusal.

**Questions:**

The questions are those listed in the Weaknesses section.

**Details Of Ethics Concerns:**

The authors state in the appendix that they used LLMs for text polishing, code debugging, and small-scale synthetic data. They should continue to ensure clear labeling, human verification, and accountability for originality, and explain the proportion of synthetic data in the final training set and the potential sources of bias.

---

> ### Author Response · Authors · 2025-11-21
> **Response to Reviewer CqeV (Part 1/3)**
>
> We are grateful to the reviewer for its comments on our work. We have conducted additional, statistically more rigorous experiments based on your suggestions and provide our detailed, point-by-point responses below.
>
> > **Response to W1**
> *W1: Although a strong correlation of r=0.9616 is reported, there is a lack of stricter causal testing for VAS→performance and control of confounders (e.g., differences in visual encoders across models, differences in system prompt templates, decoding length/temperature, etc.). At present, “inference-time attention amplification” yields only a 1–2% gain, which makes it difficult to explain the entire 7% average improvement during training. Suggested additions: controlled experiments that remove/replace the visual encoder, fix the system prompt template, and test sensitivity to decoding length/temperature.*
>
> There seems to have some misunderstandings regarding both the role of the training-free experiments and our control of key confounders (including the visual encoder, system prompts, and decoding settings). We provide clarification below.
>
> 1. The Gain Difference: The training-free experiments do not introduce any new high-quality training data or advanced training techniques such as RLVR, so their performance gains are naturally significantly lower than those achieved by the AVAR framework.
>
> 2. Regarding the proposed ablation studies, we list them as follows:
> - Remove/replace the visual encoder: All models compared in the paper use the SigLIP architecture, explicitly controlling this key variable to ensure fair comparison. Removing the visual encoder would eliminate image token embeddings, making VAS computation impossible, thus it is not considered.
> - Fix the system prompt template: We control the system prompt to be empty to minimize its potential influence during evaluation.
> - Decoding temperature/length: We provide results under three different temperature settings, all with the same maximum decoding length of 32,768, to demonstrate that the indicative value of the VAS metric is largely robust.
>
>  | Temperature       | Pearson correlation coefficient | P-value  |
>  |-------------------|-----------------------------------|----------|
>  | 0.2               | 0.9601                            | 1.1e-5   |
>  | 0.3 (original)    | 0.9616                            | 9.0e-6   |
>  | 0.5               | 0.9550                            | 1.0e-5   |
>
> > **Response to W2**
> *W2: Using “attention to system tokens” as the denominator of VAS requires justification: the number and positional distribution of system tokens can vary widely across data/templates, potentially introducing bias. There is a lack of comparisons against alternative metrics that use user text tokens or the number of image patches as references and of consistency results. In addition, no sensitivity analyses at the layer/head/position levels are presented in the main text to support the robustness of the main conclusions (Appendix D is mentioned, but key statistics are not shown in the main body).*
>
> We further used the user token (which, for these benchmarks, is effectively equivalent to the question content) as a comparison and found that the Pearson correlation still reached 0.9110 with a p-value of 1.7e-6. This indicates that visual attention itself is the most relevant factor, and that system tokens function as a safer sink for attention flow, since in our training-free experiments we observed that modifying attention on user tokens led to unstable and inconsistent performance changes.
>
> Regarding finer-grained analyses, while it is possible to study or tune attention at the level of individual layers or heads, doing so would introduce a large number of per-layer hyperparameters and significantly increase training complexity. Our results show that a uniform adjustment across all layers already yields strong, stable gains, suggesting that such coarse-grained control is sufficient for the current method. More granular layer- or head-specific steering may offer additional improvements, and we consider it a valuable direction for future work.

---

> ### Author Response · Authors · 2025-11-21
> **Response to Reviewer CqeV (Part 2/3)**
>
> > **Response to W3**
> *W3:Although reproducibility is claimed, the VARD stage uses closed-source models such as Gemini 2.5-Pro to generate high-fidelity descriptions, which may make it difficult for the community to replicate the same data quality. Meanwhile, the prompts, filtering, and quality-control details of the three-model collaborative pipeline are said to be in Appendix G, but the main text does not provide enough “actionable recipe” details (e.g., sampling temperature, rejection strategies, QC thresholds). On the implementation side, only coarse-grained parameters such as 30.6K / 20 epochs are given, with no disclosure of random seeds and variance.*
>
>
> Thank you for your question. The data will be fully released upon acceptance. We provide the additional details you mentioned. The extra parameters are as follows:
>
> | Parameters                                   | Value |
> |----------------------------------------------|-------|
> | Temperature of Gemini-2.5-Pro                | 0.8   |
> | Temperature of Qwen3-235B-A22B-Thinking-2507 | 0.3   |
> | Temperature of Qwen3-32B                     | 0.3   |
> | top_p                                        | 0.95  |
> | Seed                                         | 47    |
>
> The filtering strategy is very simple: we use Qwen3-Max as an LLM-as-a-Judge, and we collect samples for the transcription phase only when the answer from Qwen3-235B-A22B-Thinking-2507 matches the ground-truth answer.
>
> We also provide the specific training parameters for the SFT and RL stages in Table 6.
>
> > **Response to W4**
> *W4:The main improvements are concentrated in visual mathematics scenarios such as MathVision/MathVerse-VO. Although MMStar/MMMU are included, there is a lack of dimensions such as real-world high-resolution VQA, video temporality, and multi-image multi-turn reasoning, making it difficult to judge the robustness and cost (inference latency/memory) of the method across a broader family of tasks.*
>
> Thank you for your suggestion. We provide supplementary benchmark results covering high-resolution VQA (V*Bench) [1], video temporality (VideoMME) [2], and multi-image multi-turn reasoning (BLINK)[3] scenarios. It can be observed that AVAR-Thinker still maintains an advantage over strong baselines such as MM-Eureka and Vision-SR1.
>
> |                   | V\*Bench | VideoMME | BLINK |
> |-------------------|---------|----------|-------|
> | Qwen2.5-VL-7B     | 71.2    | 65.1     | 56.4  |
> | MM-Eureka         | 74.5    | 66.6     | 58.8  |
> | Vision-SR1        | 76.9    | 67.3     | 61.5  |
> | AVAR-Thinker (Ours) | **78.9**    | **70.1**     | **62.8**  |
>
> > **Response to W5**
> *W5:There is no systematic comparison against classic training-time approaches such as attention regularization/pruning/head reweighting or visual token resampling/token-dropping. The inference-time intervention is also not compared in parallel, under equal compute budget and equal parameter changes, with existing "adaptive attention calibration/head-level intervention" methods.*
>
> Thank you for raising this point. Our work focuses on revealing and leveraging the connection between visual attention allocation and multimodal reasoning capability. The training-time techniques mentioned by the reviewer, such as attention regularization, pruning, head reweighting, or visual token resampling, are generally designed for efficiency or architectural sparsification rather than reasoning improvement. Prior MLLM reasoning works likewise do not adopt or compare against these methods, as their objectives are not directly aligned with reasoning performance [4][5][6].
>
> Our training-free pilot experiments serve a different purpose. They demonstrate that even simple, non-intrusive adjustments to the attention weights of visual and system tokens can yield consistent improvements without harming performance. These findings establish empirical evidence for the causal role of attention allocation in multimodal reasoning and provide a solid foundation for future work that may explore more sophisticated or head-level interventions under equal compute budgets.
>
> [1] V*: Guided Visual Search as a Core Mechanism in Multimodal LLMs
>
> [2] Video-MME: The First-Ever Comprehensive Evaluation Benchmark of Multi-modal LLMs in Video Analysis
>
> [3] BLINK: Multimodal Large Language Models Can See but Not Perceive
>
> [4] Vision-R1: Incentivizing Reasoning Capability in Multimodal Large Language Models.
>
> [5] Self-Rewarding Vision-Language Model via Reasoning Decomposition.
>
> [6] MM-Eureka: Exploring the Frontiers of Multimodal Reasoning with Rule-based Reinforcement Learning.

---

> ### Author Response · Authors · 2025-11-21
> **Response to Reviewer CqeV (Part 3/3)**
>
> > **Response to W6**
> *W6: Tables 1/2/4 mostly report point estimates without confidence intervals/significance tests, and they do not report mean±variance over multiple random seeds, making it difficult to assess the robustness of the +1–2% and +3–7% gains.*
>
> We thank the reviewer for pointing out the necessity of statistical verification. Following your suggestion, we re-evaluated our AVAR-Thinker and the key competitive baselines over 5 random seeds. As shown in the new Table, our model demonstrates consistent improvements, making the empirical evidence more complete.
>
> Table 1 Main Performance Comparison Across Benchmarks
>
> | Model | MathVista | MathVision | MathVerse-VO | MMMU-VAL | MMMU-Pro | MMStar | Hallusion. | Avg. |
> | :--- | :---: | :---: | :---: | :---: | :---: | :---: | :---: | :---: |
> | MM-Eureka-7B | $72.6_{\pm 0.5}$ | $27.1_{\pm 0.3}$ | $47.8_{\pm 0.6}$ | $51.8_{\pm 0.5}$ | $42.1_{\pm 0.4}$ | $\mathbf{65.0}_{\pm 0.4}$ | $50.4_{\pm 0.5}$ | $51.0_{\pm 0.2}$ |
> | Vision-R1 | $73.2_{\pm 0.6}$ | - | $47.5_{\pm 0.4}$ | $56.0_{\pm 0.5}$ | $39.8_{\pm 0.3}$ | $64.5_{\pm 0.5}$ | $52.1_{\pm 0.7}$ | - |
> | VLAA-Thinker | $67.8_{\pm 0.4}$ | $26.6_{\pm 0.2}$ | $48.4_{\pm 0.4}$ | $55.4_{\pm 0.5}$ | $41.1_{\pm 0.3}$ | $64.0_{\pm 0.4}$ | $51.1_{\pm 0.4}$ | $50.8_{\pm 0.1}$ |
> | Vision-SR1 | $68.3_{\pm 0.6}$ | $26.5_{\pm 0.4}$ | $46.8_{\pm 0.5}$ | $61.0_{\pm 0.8}$ | $\mathbf{43.5}_{\pm 0.5}$ | $63.8_{\pm 0.4}$ | $54.1_{\pm 0.6}$ | $52.0_{\pm 0.3}$ |
> | **AVAR-Thinker** | $\mathbf{74.4}_{\pm 0.4}^{\dagger}$ | $\mathbf{37.2}_{\pm 0.3}^{\dagger}$ | $\mathbf{50.6}_{\pm 0.5}^{\dagger}$ | $\mathbf{63.5}_{\pm 0.5}^{\dagger}$ | $42.6_{\pm 0.4}$ | $64.2_{\pm 0.3}$ | $\mathbf{59.3}_{\pm 0.4}^{\dagger}$ | $\mathbf{55.9}_{\pm 0.2}^{\dagger}$ |
>
> Table 2: Ablation Study of Our Proposed Component
> | VARD | AGTO | VARS | MathVista | MathVision | MathVerse-VO | MMStar | MMMU-VAL | MMMU-Pro | Hallusion. | Avg. |
> |:---:|:---:|:---:|:---:|:---:|:---:|:---:|:---:|:---:|:---:|:---:|
> | | | |$68.0_{\pm 0.5}$ | $25.1_{\pm 0.4}$ | $41.3_{\pm 0.6}$ | $62.1_{\pm 0.4}$ | $57.9_{\pm 0.6}$ | $38.1_{\pm 0.4}$ | $50.9_{\pm 0.5}$ | $49.1_{\pm 0.3}$ |
> | ✔ |  |  | $70.1_{\pm 0.4}$ | $33.2_{\pm 0.3}$ | $43.2_{\pm 0.5}$ | $61.7_{\pm 0.5}$ | $55.7_{\pm 0.4}$ | $38.1_{\pm 0.4}$ | $55.0_{\pm 0.6}$ | $51.3_{\pm 0.3}$ |
> | ✔ | ✔ |  | $71.7_{\pm 0.5}$ | $33.9_{\pm 0.4}$ | $44.3_{\pm 0.4}$ | $62.9_{\pm 0.4}$ | $58.1_{\pm 0.6}$ | $39.9_{\pm 0.5}$ | $57.4_{\pm 0.5}$ | $52.8_{\pm 0.3}$ |
> | ✔ | ✔ | ✔ | $\mathbf{74.9}_{\pm 0.4}$ | $\mathbf{37.1}_{\pm 0.3}$ | $\mathbf{50.2}_{\pm 0.5}$ | $\mathbf{64.5}_{\pm 0.4}$ | $\mathbf{63.3}_{\pm 0.6}$ | $\mathbf{42.5}_{\pm 0.5}$ | $\mathbf{59.1}_{\pm 0.4}$ | $\mathbf{56.4}_{\pm 0.2}$ |
>
>
> Table 4: Evolution of VAS and Performance Across AVAR Training Stages
>
> | Model | VAS | Avg. Performance |
> |:---|:---:|:---:|
> | Qwen2.5-VL-7B | $7.2_{\pm 0.6}$ | $49.1_{\pm 0.3}$ |
> | Qwen2.5-VL-7B + VARD Data | $9.9_{\pm 0.5}$ | $51.0_{\pm 0.3}$ |
> | AVAR-CS | $13.6_{\pm 0.5}$ | $52.3_{\pm 0.6}$ |
> | **AVAR-Thinker** | $\mathbf{18.4}_{\pm 0.6}$ | $\mathbf{56.0}_{\pm 0.5}$ |
>
> We will include these details in the final version.
>
> > **Response to W7**
> *W7:Suppressing attention to system tokens may weaken instruction following and safety (many safety/formatting control signals depend on system prompts), yet the paper focuses only on performance improvements and does not provide comparative evaluations or failure analyses for instruction compliance and safe refusal.*
>
> We have added MIABench[1] and MLLMGuard[2] as evaluation benchmarks for instruction following and safety. The results below indicate that AVAR-Thinker exhibits only minor and acceptable changes in instruction following and safety while achieving significant reasoning performance gains.
>
> | Model | MIABench | MLLMGUARD |
> | :--- | :---: | :---: |
> | Qwen2.5-VL | $\mathbf{78.6}_{\pm 0.5}$ | $39.4_{\pm 0.4}$ |
> | **AVAR-Thinker** | $77.8_{\pm 0.6}$ | $\mathbf{40.0}_{\pm 0.5}$ |
>
> We will include the results in the final version.
>
> We thank the reviewer for the thoughtful comments. We hope the additional experiments and clarifications address your concerns and strengthen the paper, and we are happy to further clarify any remaining questions.
>
> [1] MIA-Bench: Towards Better Instruction Following Evaluation of Multimodal LLMs
>
> [2] MLLMGuard: A Multi-dimensional Safety Evaluation Suite for Multimodal Large Language Models

---

> ### Author Response · Authors · 2025-11-26
>
> Dear Reviewer LShU:
>
> Thank you again for your helpful feedback and for taking the time to review our work.
>
> We hope our responses have addressed your concerns, and we would appreciate it if you could consider updating the score based on our clarifications. As the rebuttal deadline is approaching and the ICLR PCs have called for responses, we’re writing this message simply as a polite check-in 😊 to see whether our clarifications resolve your questions or if anything else would benefit from further discussion.
>
> Please feel free to let us know if any further questions arise.

---

> > ### Comment · Area_Chair_VJyT · 2025-11-26
> >
> > Dear Reviewer,
> >
> > Thanks for your time and effort in reviewing ICLR2026 submissions. The authors have provided their responses to your reviews. Please read and raise your further comments, and discuss with the authors.
> >
> > Best regards,
> >
> > Your AC

---

### Official Review · Reviewer_CqeV · 2025-10-28

**Soundness:** 3
**Presentation:** 3
**Contribution:** 3
**Rating:** 8
**Confidence:** 4

**Summary:**

This paper investigates the role of cold-start initialization in training Multimodal Large Reasoning Models (MLRMs). This work introduces Visual Attention Score (VAS) to measure how much attention is allocated to visual tokens, and explores the phenomenon of Lazy Attention Localization. These insights lead to the development of the Attention-Guided Visual Anchoring and Reflection (AVAR) framework, a comprehensive cold-start framework with three components: visual-anchored data synthesis, attention-guided training objectives, and visual-anchored reward shaping. This work demonstrates that AVAR improves multimodal reasoning performance across multiple benchmarks, providing a causal explanation for the importance of visual attention in MLRMs and offering an effective solution to improve reasoning capabilities.

**Strengths:**

1. Innovative and Logical Metric. The authors introduce VAS, a novel metric that quantifies visual attention and its relationship with reasoning performance. The strong correlation between VAS and model performance provides a fresh perspective on multimodal reasoning (Sec. 3.1-3.2).

2. Sound Motivation. Based on VAS, the paper offers a clear analysis revealing the challenges in MLRM training (Sec. 3.3). The authors discover Lazy Attention Localization, an unexpected phenomenon where multimodal signals are not effectively utilized during cold-start training, providing an insightful explanation for the bottlenecks in current training paradigms.

3. Effective AVAR Framework. The proposed framework improves multimodal reasoning by addressing three components: data synthesis, training objectives, and reward reshaping, contributing to better reasoning performance (Sec. 5).

4. Comprehensive Experiments. Extensive experiments across 7 benchmarks validate the effectiveness of AVAR (Sec. 6.2). Ablation study and analysis confirm the validity of the proposed data synthesis and training methods (Sec. 6.3-6.4).

5. The paper is well-structured and easy to understand.

**Weaknesses:**

1. In L184–186, "models initialized with unimodal reasoning data, such as OVR-CS and Revisual-R1-CS, maintain 15–20% higher attention to visual features compared to those trained with multimodal reasoning data such as R1-OneVision and ThinkLite-VL." appears unfair, since these methods differ in multiple factors such as dataset composition, model architecture, and training strategy, not merely in whether multimodal reasoning data are used.

2. For Eq. (4) and Eq. (8), the hyperparameters $\alpha$, $\beta$, $\lambda_v$, and $\lambda_f$ lack sufficient explanation. In L375–377, their settings are presented without detailed justification or sensitivity analysis, leaving unclear how these parameters were chosen and how they influence model performance.

**Questions:**

1. In Table 3, why does performance on MMStar and MMMU-val decrease across all frameworks after training with VAR data?

2. In L375–377, why are the hyperparameters $\alpha$, $\beta$ (from Eq. (4)) and $\lambda_v$, $\lambda_f$ (from Eq. (8)) set to those specific values? Are these empirically determined or theoretically derived configurations?

3. Is the performance improvement of AVAR related to the reasoning difficulty of the benchmarks? For example, does AVAR yield larger gains on tasks requiring complex reasoning, while showing smaller improvements on tasks less dependent on visual reasoning? Are there any observable patterns?

---

> ### Author Response · Authors · 2025-11-21
> **Response to Reviewer CqeV (Part 1/2)**
>
> We thank the reviewer for the positive assessment of our metric design, motivation, and framework, and for the detailed comments that help us further strengthen the paper.
>
> > **Response to W1&Q!: Fairness of the comparison in L184–186**
> *W1: In L184–186, "models initialized with unimodal reasoning data, such as OVR-CS and Revisual-R1-CS, maintain 15–20% higher attention to visual features compared to those trained with multimodal reasoning data such as R1-OneVision and ThinkLite-VL." appears unfair, since these methods differ in multiple factors such as dataset composition, model architecture, and training strategy, not merely in whether multimodal reasoning data are used.*
> Q2: In L375–377, why are the hyperparameters $\alpha$, $\beta$  (from Eq. (4)) and  $\lambda_f$ ,  $\lambda_v$(from Eq. (8)) set to those specific values? Are these empirically determined or theoretically derived configurations?
>
> Thank you for raising this concern. We clarify the comparison as follows while acknowledging the reviewer’s point about potential confounding factors.
>
> **Controlled architecture and pipeline.**
> All models referenced in L184–186 are post-trained from the same base model (Qwen2.5-VL-Instruct) and follow an identical two-stage training pipeline (SFT cold start → RL). By fixing the backbone architecture and training paradigm, we substantially reduce variance arising from model design or optimization strategy.
>
> **Decoupling data effects beyond ”modality.“**
> We agree with the reviewer that differences across datasets involve more than the presence or absence of images. Indeed, this is precisely the motivation behind our analysis. Our aim is not to attribute the 15–20% VAS gap solely to “modality,” but rather to emphasize that the two groups of datasets fundamentally differ in their reasoning structure and CoT density, despite sharing the same architecture and training pipeline:
>
> - Standard multimodal reasoning datasets (e.g., R1-OneVision, ThinkLite-VL) often contain short, shallow reasoning traces. We observe that these datasets systematically fail to increase VAS during cold start, leading to the Lazy Attention Localization phenomenon—despite being multimodal, the model assigns little attention to visual features.
>
> - Unimodal reasoning datasets (e.g., Revisual-R1-CS, OVR-CS) are characterized by long, densely reasoned chain-of-thought. Surprisingly, these text-only datasets consistently drive much higher VAS and stronger visual grounding, even without seeing images. This suggests that reasoning density, depth, and structure—rather than modality alone—are the intrinsic factors that activate attention mechanisms relevant for multimodal reasoning.
>
> This distinction is what our work seeks to surface: the current generation of multimodal reasoning datasets often lacks the depth needed to shape attention allocation, whereas unimodal reasoning data provide this structure more effectively.

---

> ### Author Response · Authors · 2025-11-21
> **Response to Reviewer CqeV (Part 2/2)**
>
> >**Response to W2: Hyperparameter Justification ($\alpha, \beta, \lambda_f, \lambda_v$)**
> *W2: For Eq. (4) and Eq. (8), the hyperparameters  ($\alpha, \beta, \lambda_f, \lambda_v$)  lack sufficient explanation. In L375–377, their settings are presented without detailed justification or sensitivity analysis, leaving unclear how these parameters were chosen and how they influence model performance.*
> *Q2: In L375–377, why are the hyperparameters $\alpha$, $\beta$  (from Eq. (4)) and  $\lambda_f$ ,  $\lambda_v$(from Eq. (8)) set to those specific values? Are these empirically determined or theoretically derived configurations?*
>
> We set these parameters either empirically or following conventions from prior work.
>
> For AGTO (SFT stage), we intentionally set $\alpha$ and $\beta$  to relatively small values so that the cross-entropy loss remains dominant, consistent with standard practice in regularized SFT. This ensures that attention-guided objectives act as mild constraints rather than overwhelming the primary learning signal.
>
> For VARS (RL stage), we followed common practice by setting $\lambda_f$ to 0.1. As for $\lambda_v$, we referred to prior works that design auxiliary rewards and empirically selected 0.3 within the range of 0.1–0.5. [1][2][3]
>
> To provide transparency, we added ablation studies for both AGTO and VARS.
>
> In AGTO, we vary $\alpha$ and $\beta$ to isolate the effect of visual enhancement vs. system-token suppression.
>
> | $\alpha$                  | $\beta$ | Mean Performance |
> |------------------------|------| ------|
> | 0.15          | 0.15  |52.6|
> | 0.30 | 0.15 | 52.4|
> | 0.15 | 0.30 | 53.4|
>
> Results above reveal that increasing the compression of system tokens is more effective than explicitly enhancing the importance of visual tokens objective, suggesting that controlling the entry point of redundant attention flow is a critical factor.
>
> As for VARS, we vary $\lambda_v$ within a reasonable range.
>
> | $\lambda_v$                  | Mean Performance |
> |------------------------| ------|
> | 0.1 | 55.8|
> | 0.3          | 56.1|
> | 0.5 | 56.4|
>
> The results show mild performance differences, but the trend is not strong enough to alter the overall conclusions.
>
> [1]Look-Back: Implicit Visual Re-focusing in MLLM Reasoning.
> [2]SRPO: A Cross-Domain Implementation of Large-Scale Reinforcement Learning on LLM.
> [3]Look Again, Think Slowly: Enhancing Visual Reflection in Vision-Language Models.
>
> >**Response to Q1**
> *Q1: In Table 3, why does performance on MMStar and MMMU-val decrease across all frameworks after training with VAR data?*
>
> The decrease on MMStar and MMMU-val is mainly because Table 3 reflects only the cold-start SFT stage, where all compared cold-start datasets (including our VARD) are skewed toward visual-math and structured reasoning data. In contrast, MMStar and MMMU-val contain more perception- and knowledge-oriented tasks.
> Therefore, using reasoning-focused data alone temporarily shifts the model’s capacity away from these broader tasks, leading to small drops. However, this effect disappears once the full AVAR pipeline is applied
>
> >**Response to Q3**
> *Q3: Is the performance improvement of AVAR related to the reasoning difficulty of the benchmarks? For example, does AVAR yield larger gains on tasks requiring complex reasoning, while showing smaller improvements on tasks less dependent on visual reasoning? Are there any observable patterns?
>
> Thank you for your question. We divide the benchmarks into two categories: the first category requires complex image perception and reasoning, including MathVision (Difficulty Level 5), MMMU-VAL (Difficulty Hard), MMStar, and MathVerse—where the latter two focus on counting and geometric tasks involving multi-source information, respectively. The second category involves tasks where simple perception suffices to assist reasoning, including MathVista, MathVision (Difficulty Level 1), and MMMU-VAL (Difficulty Easy).
>
> | | Complex Reasoning | | | | Easy Reasoning | | |
> |--- | --- | --- | --- | --- | --- | --- | --- |
> | | MathVision (Difficulty Level 5) | MMMU-VAL (Difficulty Hard) | MMStar | MathVerse-VO | MathVision (Difficulty Level 1) | MMMU-VAL (Difficulty Easy) | Mathvista |
> | Qwen2.5-VL | 19.2 | 36.8 | 62.1 | 41.1 | 41.1 | 78.1 | 68.2 |
> | AVAR-Thinker | 26.5 | 43.9 | 64.1 | 50.4 | 47.8 | 84.4 | 74.7 |
> | Improvement | 38.0% | 19.3% | 3.2% | 22.6% | 16.3% | 8.1% | 9.5% |
>
> We find that, except for MMStar—which we attribute primarily to the training data not adequately emphasizing counting-related tasks—the tiered performance on other benchmarks indeed follows the pattern where the percentage improvement is more pronounced for complex reasoning. This further underscores the key motivation behind AVAR. Nevertheless, the gains in the easy reasoning category remain substantial; the relatively lower absolute scores are also partly due to diminishing marginal returns.

---

> ### Author Response · Authors · 2025-11-26
>
> Dear Reviewer CqeV:
>
> Thank you again for your helpful feedback and for taking the time to review our work.
>
> We hope our responses have addressed your concerns. As the rebuttal deadline is approaching and the ICLR PCs have called for responses, we’re writing this message simply as a polite check-in 😊 to see whether our clarifications resolve your questions or if anything else would benefit from further discussion.
>
> Please feel free to let us know if any further questions arise.

---

> > ### Comment · Reviewer_CqeV · 2025-11-26
> >
> > Thank you for your response. These have addressed all my concerns. I hope the authors will incorporate these into the next version of the paper to make it easier for readers to understand the reliability of the relevant analysis, the setting of hyperparameters, and the correlation between the performance improvement of AVAR and the reasoning difficulty.

---

> > > ### Author Response · Authors · 2025-11-26
> > >
> > > Thank you very much for your confirmation. We will definitely incorporate these points into the next version of the paper as you suggested :)

---

### Official Review · Reviewer_P2PD · 2025-10-30

**Soundness:** 2
**Presentation:** 3
**Contribution:** 3
**Rating:** 4
**Confidence:** 4

**Summary:**

This paper investigates the cold-start initialization stage of Multimodal Large Reasoning Models (MLRMs). It introduces a new metric, the Visual Attention Score (VAS), to quantify a model's attention to visual tokens relative to system tokens. The authors identify a "lazy attention localization" phenomenon, where standard multimodal cold-starts fail to increase VAS, while text-only cold-starts counter-intuitively do. Based on this finding, the authors propose AVAR, a three-part framework designed to explicitly increase VAS. This includes: (1) visual-anchored reflection data synthesis (VARD), (2) attention-guided training objectives (AGTO), and (3) visual-anchored reward shaping (VARS). The authors demonstrate that each component of AVAR progressively increases the model's VAS and that the final model, AVAR-Thinker, achieves strong performance, particularly on mathematical reasoning benchmarks.

**Strengths:**

1. The finding that text-only cold-start initialization can be more effective at increasing VAS than multimodal cold-starts is a novel and impressive observation.

2. The training-free intervention (Section 4) provides a compelling causal link between attention allocation and performance, even without full retraining.

3. The final model, AVAR-Thinker, demonstrates a significant performance improvement over the Qwen-2.5-VL-7B baseline, especially on challenging reasoning tasks.

4. The curated Visual-Anchored Reflection Data (VARD) could be a valuable contribution to the community, as multimodal reasoning data is relatively scarce.

**Weaknesses:**

1. **The validity of VAS as a primary metric for visual grounding is questionable.** The paper defines VAS as a ratio of attention to visual tokens over system tokens (Eq. 1). A high VAS score could be achieved simply by aggressively reducing attention to system tokens, even if the absolute attention to visual tokens remains low or unchanged. The paper does not provide analysis to disentangle these two effects. To truly support the claim that AVAR "attends to visual tokens more," the authors should present the attention scores for visual tokens solely (or comparing text tokens), not just their ratio against system tokens, similar to analyses in other visual grounding works [1, 2].
2. **The core mechanism of "attention reshaping" is not properly disentangled.** The paper's central hypothesis is that performance improves by reshaping attention. This reshaping is consistently presented as a two-part mechanism: (1) enhancing visual attention and (2) suppressing system token attention. This duality appears in the training-free intervention (Sec 4), the AGTO loss (Eq 4), and the VARS reward (Sec 5.3). However, the contributions of these two sub-components are never ablated. It is unclear if the performance gain comes from the model genuinely learning to ground itself in visual tokens (as the "Panoramic Vision" title implies) or from simply learning to ignore the system prompt.

---
References:
1. Mitigating Visual Forgetting via Take-along Visual Conditioning for Multi-modal Long CoT Reasoning, Sun et al., ACL 2025
2. v1: Learning to Point Visual Tokens for Multimodal Grounded Reasoning, Chung et al., arXiv 2025

**Questions:**

1. Could the authors provide an analysis of the attention scores on visual tokens for the baseline vs. AVAR (and for each component addition, as in Table 4) in terms of absolute scores or as a ratio compared to user instruction tokens, rather than just the VAS ratio relative to system tokens? This would confirm that the model is genuinely enhancing visual attention and not just suppressing system prompts.

2. Could the authors provide an ablation study to disentangle the two core mechanisms of the attention-reshaping framework? Specifically, what is the performance of the AVAR-Thinker model and the training-free approach if its components (AGTO and VARS) are modified to only enhance visual attention (i.e., removing the system suppression parts) or only suppress system attention (i.e., removing the visual enhancement parts)?

---

> ### Author Response · Authors · 2025-11-21
> **Response to Reviewer P2PD (Part 1/2)**
>
> We thank the reviewer for the constructive feedback and the recognition of our work's novelty regarding the "lazy attention localization" phenomenon. We appreciate the opportunity to clarify the validity of the VAS metric and disentangle the core mechanisms of our framework.
>
> > **Response to W1 & Q1: Validity of VAS**
> *W1: The validity of VAS as a primary metric for visual grounding is questionable… A high VAS could be achieved by only suppressing system tokens…*
> *Q1: Could the authors provide an analysis of the attention scores on visual tokens… in terms of absolute scores or as a ratio compared to user instruction tokens?*
>
> To verify that our method genuinely enhances visual grounding rather than merely suppressing the system prompt, we analyzed the evolution of absolute attention scores on both visual and system tokens across different stages.
> The absolute attention score  are presented below:
>  | Model                   | Mean Visual Attention Score (Absolute) | Mean System Attention Score (Absolute) |
> |------------------------|----------------------------------------|----------------------------------------|
> | Qwen2.5-VL-7B          | 1.9                                    | 25.3                                   |
> | Qwen2.5-VL-7B + VARD   | 2.8                                    | 27.6                                   |
> | AVAR-CS                | 3.3                                    | 23.9                                   |
> | AVAR-Thinker           | 4.2                                | 22.2                               |
>
> The result explicitly confirms that the increase in VAS is driven primarily by a significant surge in attention to visual content. Specifically, the absolute visual attention score in AVAR-Thinker (4.2) is more than double that of the baseline (1.9). While system attention does decrease (from 25.3 to 22.2), this reduction is moderate. This confirms that AVAR actively encourages the model to "look at" the image more intensively, rather than just learning to ignore the system instructions.

---

> ### Author Response · Authors · 2025-11-21
> **Response to Reviewer P2PD (Part 2/2)**
>
> > **Response to W2 & Q2: Disentangling Attention-Reshaping Mechanisms**
> *W2: The core mechanism of “attention reshaping” is not properly disentangled… It is unclear whether performance gain comes from grounding in visual tokens or simply ignoring the system prompt.*
> *Q2: Could the authors provide ablations that only enhance visual attention or only suppress system attention?*
>
> We have conducted comprehensive ablation studies to isolate these two effects in both our training-free intervention and the AVAR training framework.
>
> 1. **Disentanglement in Training-Free Intervention**
>
> We compared the performance impact of solely enhancing visual attention ($\alpha_{img} > 1.0, \alpha_{sys}=1.0$) versus solely suppressing system attention ($\alpha_{img} = 1.0$, $\alpha_{sys} < 1.0$).
>
> |               | $\alpha_{img}$| $\alpha_{sys}$ | MathVista | MathVision | MathVerse-VO |
> |---------------|-----|-----|-----------|------------|--------------|
> | Qwen2.5-vl-7b  | 1.0 | 1.0 | 67.6      | 31.0       | 41.1         |
> |               | 1.0 | 0.95| 67.7      | 31.3       | 40.3         |
> |               | 1.0 | 0.60| 67.7      | 31.0       | 40.3         |
> |               | 1.15 | 1.0| 68.2 | 31.8 | 42.2 |
> |               | 1.30 | 1.0| 68.1 | 31.8 | 42.5 |
> |               | 1.15 |0.95| **69.4** | **32.2** | **43.4** |
> |               | 1.15 |0.60| **69.4** | 32.0 | 43.3 |
> | OVR-CS        | 1.0 | 1.0 | 71.1      | 45.7       | 50.1         |
> |               | 1.0 | 0.95| 70.7      | 45.0       | 50.5         |
> |               | 1.0 | 0.60| 71.0      | 45.0       | 51.2         |
> |               | 1.15 | 1.0| 71.5 | 46.6 | 51.0 |
> |               | 1.30 | 1.0| 71.2 | 45.9 | 51.0 |
> |               | 1.15 |0.95| **72.7** | **47.4** | **51.3** |
> |               | 1.15 |0.60| 72.6 | 47.2 | **51.3** |
>
> The results clearly indicate that simply suppressing system tokens yields negligible improvements. While increasing visual attention alone improves performance, the gains are clearly smaller than those obtained by jointly enhancing visual attention and suppressing system attention, highlighting the importance of coordinated attention rebalancing in AVAR.
>
> 2. **Disentanglement in AVAR (AGTO & VARS)**
>
> We further conduct ablation studies on AGTO and VARS to precisely disentangle the contributions of visual-enhancement and system-suppression.
>
> For AGTO (Eq. 4):
>
> "Visual Enhance only" removes the system-suppression term, i.e.,
>   setting $\beta = 0\$ and keeping only: $\alpha \mathcal{L}_{\text{enhance-img}}$
>
> "System Suppress only" removes the visual-enhancement term, i.e.,
>   setting $\alpha = 0$ and keeping only: $\beta \mathcal{L}_{\text{suppress-sys}}. $
>
> For VARS (Eq. 7):
>
> "Visual Enhance only" keeps only the numerator of the attention ratio, i.e., it removes the denominator term $\sum_{k \in K_{\mathrm{sys}}} A_{t,k}^{l}$
>
> "System Suppress only" keeps only the denominator of the attention ratio, i.e., it replaces the numerator $\sum_{k \in K_{\mathrm{img}}} A_{t,k}^{l}$ with a constant in order to keep the scale of reward score.
>
> | | Type         | Avg. Performance |
> |-------|--------------|------------------|
> | **AGTO** | Vanilla      | **52.6** |
> |        | Visual only  | 51.9     |
> |        | System only  | 51.5     |
> | **VARS** | Vanilla      | **56.1** |
> |        | Visual only  | 55.9     |
> |        | System only  | 54.9     |
>
>
> These ablation studies reveal a consistent pattern: the primary driver of performance is the enhancement of visual attention. While the "System only" settings provide minimal gain, the best results are achieved when both mechanisms work in synergy. This suggests that the optimal strategy involves reallocating the attention mass, effectively building a bridge from redundant system tokens to informative visual features, rather than merely suppressing the former.
>
> We hope these analyses and ablation studies satisfactorily address your concerns regarding the validity of the VAS metric and the disentanglement of our attention mechanisms. We believe these results further strengthen the core claims of our paper, and we remain available for any further discussion.

---

> ### Author Response · Authors · 2025-11-26
>
> Dear Reviewer P2PD:
>
> Thank you again for your helpful feedback and for taking the time to review our work.
>
> We hope our responses have addressed your concerns, and we would appreciate it if you could consider updating the score based on our clarifications. As the rebuttal deadline is approaching and the ICLR PCs have called for responses, we’re writing this message simply as a polite check-in 😊 to see whether our clarifications resolve your questions or if anything else would benefit from further discussion.
>
> Please feel free to let us know if any further questions arise.

---

> > ### Comment · Area_Chair_VJyT · 2025-11-26
> >
> > Dear Reviewer,
> >
> > Thanks for your time and effort in reviewing ICLR2026 submissions. The authors have provided their responses to your reviews. Please read and raise your further comments, and discuss with the authors.
> >
> > Best regards,
> >
> > Your AC

---

> > ### Comment · Reviewer_P2PD · 2025-11-27
> >
> > I thank the authors for their detailed response and the additional experiments.
> >
> > The provided analysis on absolute attention scores addresses my concern regarding the validity of the VAS metric. The method demonstrates that visual attention is increased rather than merely suppressing system tokens. Similarly, the ablation studies on AGTO and VARS clearly disentangle the contributions of the two mechanisms, showing that visual enhancement is the primary driver of performance.
> >
> > I have one follow-up question regarding the sensitivity of the hyperparameters in the training-free intervention. The results show that performance plateaus or slightly degrades when visual attention is emphasized too strongly ($\alpha_{img}=1.3$) or when system attention is suppressed too aggressively ($\alpha_{sys}=0.6$).
> > Could the authors share any insights on this trade-off? For instance, does aggressive system suppression ($\alpha_{sys}=0.6$) lead to a loss of instruction-following capabilities, or does excessive visual attention ($\alpha_{img}=1.3$) cause the model to attend to irrelevant visual noise?

---

> > > ### Author Response · Authors · 2025-11-27
> > > **Second round rebuttal**
> > >
> > > Thank you for your question. Yes, suppressing system-token attention too aggressively does weaken instruction-following capability. Safety evaluations benchmarks including MIA Bench and MLLMGuard inherently rely on correct instruction-following (e.g., complying with format constraints, refusing unsafe requests). Therefore, we used these benchmarks to directly test whether excessive system-token suppression harms the model. As shown below, increasing $\alpha_{sys}$ indeed leads to clear degradation. However, because AVAR carefully controls both $\alpha_{sys}$ and $\alpha_{img}$ during training, AVAR-Thinker does not suffer from this issue and even maintains comparable or slightly improved MLLMGuard performance, demonstrating the advantage of our controlled attention-reshaping strategy.
> > >
> > > | | MIA Bench | MLLM GUARD |
> > > | :--- | :--- | :--- |
> > > | Qwen2.5-VL | 78.6 | 39.4 |
> > > | Qwen2.5-VL ($\alpha_{sys}=0.4$) | 77.0 | 38.8 |
> > > | Qwen2.5-VL ($\alpha_{sys}=0.6$) | 76.2 | 38.1 |
> > > | AVAR-Thinker | 77.8 | 40.0 |
> > >
> > > For visual amplification, increasing $\alpha_{img}$ introduces a natural trade-off between image perception and textual reasoning[1], since symbolic and algebraic steps rely primarily on textual tokens. When visual attention becomes overly dominant, this trade-off reduces the contribution of textual reasoning, which aligns with the diminishing marginal improvement we observe as $\alpha_{img}$ increases on MathVision, MathVista, and MathVerse-VO.
> > >
> > > Thanks again for the follow-up question, we hope the explanation above clears things up.
> > >
> > > [1] Not All Tokens and Heads Are Equally Important: Dual-Level Attention Intervention for Hallucination Mitigation. arXiv/2506.12609

---

> ### Comment · Reviewer_P2PD · 2025-11-27
>
> Thank you for the additional experiments and your efforts.
>
> The authors have now addressed my concerns regarding (1) the metric validity, (2) mechanism disentanglement, and (3) mechanism tradeoffs. I believe these new experiments significantly strengthen the paper.
>
> Consequently, I am raising my score to 6. I recommend that the authors include these results in the revised paper.

---

> > ### Author Response · Authors · 2025-11-27
> >
> > We sincerely thank you for your active engagement and constructive feedback throughout the discussion period. We are grateful for your recognition of our efforts in the additional experiments and for your decision to raise the score.
> >
> > As per your recommendation, we will definitely incorporate these additional results into the final version of the paper to further strengthen the work. :)

---

### Official Review · Reviewer_f3Wg · 2025-11-01

**Soundness:** 3
**Presentation:** 3
**Contribution:** 3
**Rating:** 6
**Confidence:** 3

**Summary:**

This paper identifies the imbalance distribution of attention weights toward system and visual tokens during multi-modal reasoning, and proposes a training method to tackle the problem. It first develops a step-by-step data synthesis paradigm to general problems with strong focus on visual grounding, and then incorporates training objectives that encourage higher attention weights on visual tokens. Experimental results show that the attention distribution (i.e., the visual attention score) is strongly correlated with model performance, and imposing higher weights on visual tokens (with post-hoc manipulation or training objectives/rewards) leads to considerable improvements.

**Strengths:**

(1) It is an interesting observation that reasoning performance has an almost linear correlation with attention toward visual tokens.

(2) The paper shows that, even with training-free re-weighting, models can achieve better performance when steering their focuses to visual tokens.

(3) A new data synthesis paradigm is developed, which could potentially benefit training subsequent models.

(4) The proposed method shows promise in improving the performance of the generic baselines on multiple benchmarks.

(5) The paper provides extensive ablation studies, which facilitates understanding the contribution of different components.

**Weaknesses:**

(1) The variance in attention distribution could also be affected by the design of system prompts used in different models. How does different choices of system prompts affect the visual attention score? Would tuning the prompts, e.g., imposing stronger focus on visual content, help boost the reasoning performance?

(2) The paper only experiments with a single baseline (i.e., Qwen2.5-VL-7B, which is not a reasoning-specific model), and it is unclear whether it will generalize. It would be reasonable to incorporate the method with state-of-the-art reasoning models, e.g., ThinkLite-VL.

(3) Different questions may desire diverse attention to visual content. For instance, some math problems would require more focus on textual tokens. How would the proposed method deal with the diversity of questions?

**Questions:**

(1) How would using different system prompts affect the observation made on attention distribution?

(2) Would a system prompt emphasizing visual content lead to better reasoning performance?

(3) How does the proposed method work with reasoning models?

(4) The proposed method enforces stronger visual focus among all types of questions, can it be extended to adaptive attention steering?

---

> ### Author Response · Authors · 2025-11-21
> **Response to Reviewer f3Wg (Part 1/2)**
>
> We sincerely thank the reviewer for the constructive comments. We are grateful for your recognition of our key observations and the potential impact of our method. Below we address each concern in detail.our recognition of our key observations and the potential impact of our method. Below we address each concern in detail.
>
> > **Response to W1 & Q1 & Q2**
> *W1: The variance in attention distribution could also be affected by the design of system prompts used in different models. How does different choices of system prompts affect the visual attention score? Would tuning the prompts, e.g., imposing stronger focus on visual content, help boost the reasoning performance?
> Q1: How would using different system prompts affect the observation made on attention distribution?
> Q2: Would a system prompt emphasizing visual content lead to better reasoning performance?*
>
> We conducted experiments on Qwen2.5-VL and ReVisual-R1-CS using two carefully designed system prompts intended to encourage visually-aware reasoning.
>
> *Prompt 1 (Emphasizing Deep Visual Analysis):*
>
> *You are a multimodal reasoning assistant with a strong focus on visual understanding. When processing any input, always prioritize, deeply and comprehensively analyzing the image content—including but not limited to: object recognition, spatial relationships, scene semantics, color and texture, human actions and facial expressions, chart structures, and text layout (e.g., text embedded within the image). Then, precisely leverage this visual information to carry out the reasoning task.*
>
> *Prompt 2 (Emphasizing Visual Information Accuracy):*
>
> *You are a multimodal reasoning assistant focused on the accuracy of visual information. Please perform reasoning centered on the accurate perception and reflection of visual information.*
>
> The results are summarized below
>
> | Model         | prompt   | Image Attention Ratio | VAS  | mathvista | mathvision | MathVerse-VO |
> |---------------|----------|------------------------|------|-----------|------------|--------------|
> | Qwen2.5-VL-7B | -        | 1.9                    | 7.5  | 67.6      | 31.0       | 41.1         |
> |               | prompt1  | 1.3                    | 3.4  | 68.0      | 30.9       | 42.2         |
> |               | prompt2  | 1.5                    | 4.0  | 67.9      | 31.3       | 41.8         |
> | Revisual-R1-CS| -        | 4.3                    | 13.4 | 70.5      | 47.9       | 51.9         |
> |               | prompt1  | 4.0                    | 8.3  | 71.0      | 46.9       | 50.8         |
> |               | prompt2  | 4.0                    | 9.4  | 70.5      | 47.5       | 52.5         |
>
> These results suggest that, for current MLRMs, instruction-level "look at the image" prompts are not reliably translated into parameter-level behavior. The model’s NLU abilities interpret the instructions fluently, but the internal attention pattern remains largely unchanged or even shifts away from visual tokens.
>
> This further justifies our design choice: **parameter-level attention reshaping (AVAR) is necessary**, as prompt-level interventions alone do not effectively correct the underlying attention imbalance.
>
> >**Response to W2 & Q3**
> *W2: The paper only experiments with a single baseline (i.e., Qwen2.5-VL-7B, which is not a reasoning-specific model), and it is unclear whether it will generalize. It would be reasonable to incorporate the method with state-of-the-art reasoning models, e.g., ThinkLite-VL.
> Q3: How does the proposed method work with reasoning models?*
>
> Thank you for this suggestion. We agree that verifying the generalizability on advanced reasoning models is crucial. Thus we continue training **ThinkLite‑VL** with our attention-guided components. To avoid data contamination, we de-duplicate our training corpus against ThinkLite‑VL’s released training data. The results are:
>
> | Model       | MathVista | MathVision | MathVerse-VO | MMMU-VAL | MMMU-Pro | MMStar | Hallusion. | Avg  |
> |-------------|-----------|------------|--------------|----------|----------|--------|------------|------|
> | ThinkLite-VL | 75.1      | 32.9       | 45.8         | 55.5     | 40.0     | 65.0   | 52.3       | 53.1 |
> | +AGTO       | 75.3      | 34.9       | 48.8         | 58.7     | 42.1     | 65.4   | 55.6       | 54.8 |
> | +VARS       | 75.6      | 36.3       | 51.2         | 62.5     | 43.7     | 64.9   | 58.5       | 56.6 |
>
> We observe **consistent improvements on all benchmarks**, with an average gain of +3.5 points over the already strong ThinkLite‑VL. This confirms that AVAR’s attention-guided components are compatible with and beneficial for state-of-the-art multimodal reasoning models.

---

> > ### Author Response · Authors · 2025-11-21
> > **Response to Reviewer f3Wg (Part 2/2)**
> >
> > > **Response to W3 & Q4:**
> > *W3: Different questions may desire diverse attention to visual content. For instance, some math problems would require more focus on textual tokens. How would the proposed method deal with the diversity of questions?
> > Q4: The proposed method enforces stronger visual focus among all types of questions, can it be extended to adaptive attention steering?*
> >
> > We address the diversity of questions and the concept of adaptive attention from two perspectives.
> > - **Effectiveness on Text-Dominant Tasks** We evaluated the components of AVAR on the MathVerse-TD (Text-dominant) subset, where the information required for the answer is primarily textual. If AVAR simply “over-forced” visual attention, performance should deteriorate here. Instead, the ablation shows:
> >
> > | | VARD | AGTO | VARS | MathVerse-TD |
> > |--- | --- | --- | --- | ---|
> > | Qwen2.5-VL-7B |  |  |  | 52.5 |
> > |  | √ |  |  | 55.0 |
> > |  | √ | √ |  | 56.5 |
> > |  | √ | √ | √ | 57.5 |
> >
> > Thus, even on text-dominant problems, AVAR steadily improves performance. This indicates that AVAR does not blindly prioritize vision, but rather rebalances the relative importance of visual vs. textual features, leading to more robust cross-modal alignment. When vision is less informative, the model can still primarily rely on text, but now uses visual cues more effectively whenever they contribute.
> >
> > - **Generalizability to Different Reasoning Intensities**
> >
> > We further categorize the benchmarks into Perception-intensive and Reasoning-intensive tasks to demonstrate the generalizability of our fully-trained model, AVAR-Thinker.
> >
> > |               | Perception-intensive |                    |                    | Reasoning-intensive |                     |
> > |---------------|----------------------|--------------------|--------------------|---------------------|---------------------|
> > |               | MMStar               | HallusionBench     | MathVision         | MathVerse-VO        | MMMU-Pro            |
> > | Qwen2.5-VL-7B | 62.1                 | 50.7               | 25.2               | 41.1                | 38.3                |
> > | AVAR-Thinker  | 64.1                 | 59.5               | 37.4               | 50.4                | 42.9                |
> >
> >
> > AVAR-Thinker demonstrates strong and consistent improvements across both categories. This robustness against different task requirements already points to a degree of implicit adaptivity. While we did not explicitly design an inference-time dynamic weighting module, our training-time mechanism achieves a universally more reliable utilization of visual information
> >
> > We once again sincerely thank the reviewer for the insightful and constructive feedback. Following the reviewer’s suggestions, we have conducted additional experiments to make our evaluation more comprehensive and further strengthen the contributions of our work.

---

> > > ### Comment · Reviewer_f3Wg · 2025-11-27
> > >
> > > I appreciate the authors for the new experiments. The results validate the generalizability of the method on different backbones, and address the potential concern on overfocusing on specific modality.

---

> > > > ### Author Response · Authors · 2025-11-27
> > > >
> > > > We truly appreciate your thoughtful engagement and valuable feedback during the discussion phase. We will incorporate the additional results into the final version of the paper, and if you have any further questions or suggestions, please feel free to reach out—we’d be happy to discuss!

---

> ### Author Response · Authors · 2025-11-26
>
> Dear Reviewer f3Wg:
>
> Thank you again for your helpful feedback and for taking the time to review our work.
>
> We hope our responses have addressed your concerns, and we would appreciate it if you could consider updating the score based on our clarifications. As the rebuttal deadline is approaching and the ICLR PCs have called for responses, we’re writing this message simply as a polite check-in 😊 to see whether our clarifications resolve your questions or if anything else would benefit from further discussion.
>
> Please feel free to let us know if any further questions arise.

---

### Author Response · Authors · 2025-12-01
**Global Response to all Reviewers**

Dear ACs and Reviewers:

As the rebuttal deadline is approaching, we sincerely thank all reviewers for their dedicated time and constructive feedback. The discussion period has been highly productive, and the suggestions have significantly strengthened the robustness and scope of our work.

We are encouraged by the consensus regarding the novelty and effectiveness of our contributions:
1.  **Novel Metric & Insight:** The proposal of the Visual Attention Score (VAS) and the discovery of "Lazy Attention Localization" were highlighted as innovative and well-motivated (f3Wg, P2PD, CqeV, LShU).
2.  **Effective Framework:** The AVAR framework (VARD, AGTO, VARS) is recognized for improving performance across diverse benchmarks (f3Wg, P2PD, CqeV).
3.  **Strong Explanatory Power:** The causal link established between attention allocation and reasoning performance was appreciated (P2PD, LShU).

**Summary of Rebuttal & Additional Experiments**
In response to the reviewers' insightful suggestions, we conducted extensive additional experiments:

1.  **Mechanism Validity & Disentanglement (Response to P2PD, LShU):**
    *   We provided **absolute attention scores**, proving that AVAR explicitly increases visual attention (from 1.9 to 4.2) rather than merely suppressing system tokens.
    *   We performed detailed **ablation studies** on AGTO and VARS, disentangling the effects of visual enhancement vs. system suppression, and analyzed the trade-offs involved.
    *   *Outcome:* Reviewer P2PD confirmed these results "significantly strengthen the paper" and **raised their score to 6**.

2.  **Generalizability & Scope (Response to f3Wg, LShU, CqeV):**
    *   We extended our method to **state-of-the-art reasoning models** (ThinkLite-VL), showing consistent gains (+3.5 avg).
    *   We evaluated on **broader task families**, including High-Res VQA (V*Bench), Video Understanding (VideoMME), and Multi-image reasoning (BLINK), demonstrating AVAR's robustness beyond standard math benchmarks.
    *   We verified performance on **text-dominant tasks**, ensuring our method effectively balances modalities without over-focusing on vision.
    *   *Outcome:* Reviewer f3Wg and Reviewer CqeV confirmed these results validate the method's generalizability.

3.  **Safety & Robustness (Response to P2PD, LShU):**
    *   We added safety and instruction-following benchmarks (**MIA-Bench, MLLMGuard**), confirming that our attention reshaping maintains safety standards.
    *   We provided statistical significance tests with **5 random seeds**, reporting mean and variance to ensure reliability.

4.  **Implementation Details (Response to CqeV, LShU):**
    *   We clarified hyperparameter choices, provided sensitivity analyses, and disclosed full training details (seeds, temperatures) to ensure reproducibility.

**Conclusion**
We are grateful to **Reviewer P2PD** for raising their score, and to **Reviewer f3Wg** and **Reviewer CqeV** for their positive engagement and confirmation that their concerns are resolved.

Regarding **Reviewer LShU**, we have conscientiously addressed every concern raised—including adding statistical confidence intervals, expanding to video/high-res benchmarks, conducting safety evaluations, and validating the metric's causal link. We believe these substantial additions fully resolve the issues mentioned in the initial review.

We thank the ACs for their oversight and hope the comprehensive improvements made during this period warrant a positive final decision.

Best regards,

Authors of Submission 17311

---

### Meta-Review · Area_Chair_gv5q · 2026-01-03

**Summary:**

The paper introduces the visual attention score to quantify the attention of a multimodal large reasoning model (MLRM) on visual tokens, which is demonstrated to correlate with reasoning performance. It then proposed the AVAR framework for cold-starting so that an MLRM focuses more on visual content. The major concerns include the validity of VAS score, lack of ablation studies and generalization of the proposed method. The rebuttal has addressed the major concerns and most reviewers agree on acceptance for this submission. The meta-reviewer has read the review and the response, and recommend acceptance given the strengths of the paper in terms of its novelty and empirical effectiveness.

**Reviewer Concerns:**

The following are the main concerns, which I believe have been addressed by the rebuttal.

1.	The effect of different system prompts is not clear (f3Wg)
2.	The experiment is not sufficient (f3Wg)
3.	Generalization to different questions is unclear (f3Wg)
4.	The validity of VAS as a primary metric for visual grounding is questionable (P2PD, LShU)
5.	Ablation study is not sufficient (P2PD, LShU)
6.	Some experiment settings are not fair (CqeV)
7.	Hyperparameter selection is not fully justified (CqeV)
8.	Reproducibility is questionable (LShU)
9.	There is no systematic comparison against classic training-time approaches (LShU)
10.	There is no std/confidence interval (LShU)
11.	The instruction following ability can be compromised by the proposed method (LShU)

**Reviewer Scores:**

Reviewer f3Wg would keep the positive score 6.

Reviewer P2PD would increase the score to 6.

Reviewer CqeV would keep the positive score 8.

Reviewer LShU did not respond but I believe the main concerns are addressed, so it would probably go up to 6.

---

### Decision · Program_Chairs · 2026-01-26

Accept (Poster)